# SUMO-Targeted Ubiquitin Ligases and Their Functions in Maintaining Genome Stability

**DOI:** 10.3390/ijms22105391

**Published:** 2021-05-20

**Authors:** Ya-Chu Chang, Marissa K. Oram, Anja-Katrin Bielinsky

**Affiliations:** Department of Biochemistry, Molecular Biology and Biophysics, University of Minnesota, Minnesota, MN 55455, USA; chan1285@umn.edu (Y.-C.C.); oramx003@umn.edu (M.K.O.)

**Keywords:** genome stability, STUbL, SUMO, ubiquitin, Slx5/Slx8, RNF4, RNF111

## Abstract

Small ubiquitin-like modifier (SUMO)-targeted E3 ubiquitin ligases (STUbLs) are specialized enzymes that recognize SUMOylated proteins and attach ubiquitin to them. They therefore connect the cellular SUMOylation and ubiquitination circuits. STUbLs participate in diverse molecular processes that span cell cycle regulated events, including DNA repair, replication, mitosis, and transcription. They operate during unperturbed conditions and in response to challenges, such as genotoxic stress. These E3 ubiquitin ligases modify their target substrates by catalyzing ubiquitin chains that form different linkages, resulting in proteolytic or non-proteolytic outcomes. Often, STUbLs function in compartmentalized environments, such as the nuclear envelope or kinetochore, and actively aid in nuclear relocalization of damaged DNA and stalled replication forks to promote DNA repair or fork restart. Furthermore, STUbLs reside in the same vicinity as SUMO proteases and deubiquitinases (DUBs), providing spatiotemporal control of their targets. In this review, we focus on the molecular mechanisms by which STUbLs help to maintain genome stability across different species.

## 1. Introduction

Genome instability is the cause of multiple syndromes that affect normal human development and can lead to uncontrolled cell proliferation, such as in cancer [1,2,3,4,5]. Cells have evolved complex surveillance mechanisms that govern cell division to prevent genomic alterations: (1) error-free and error-prone DNA repair throughout the cell cycle, (2) high-fidelity DNA replication in S-phase, (3) proper chromosome segregation in mitosis, and (4) checkpoints that coordinate cell cycle progression [2,6]. Together, these pathways ensure that the genome is faithfully replicated and evenly divided into daughter cells [2]. This review details the role of small ubiquitin-like modifier (SUMO)-targeted E3 ubiquitin ligases (STUbLs) in maintaining genomic integrity. STUbLs are evolutionarily conserved from yeast to humans. We discuss their structures and functions and highlight proteins that are direct or indirect targets of SUMO and ubiquitin regulation.

Ubiquitin and SUMO are two post-translational modifications (PTMs) with increasing prominence in genome maintenance pathways [7,8,9]. Both ubiquitin and SUMO attachments are catalyzed by similar enzyme cascades. Ubiquitin-activating enzymes (E1s) bind adenosine triphosphate (ATP) and ubiquitin to activate and transfer the moiety to ubiquitin-conjugating enzymes (E2s) through a trans-thioesterification reaction [10]. E2s bind to ubiquitin ligases (E3s), which facilitate the transfer of the ubiquitin molecule from the E2 to the designated lysine residue (K) on the target protein [10]. SUMO E1, E2, and E3s work in a similar manner to modify their substrates. However, there are notable differences between these two systems. In mammals, three active SUMO members, SUMO1-3, and two less common members, SUMO4-5, constitute the SUMO family, whereas a single ubiquitin peptide is expressed. Unlike SUMO, the ubiquitin peptide is produced as a precursor that needs to be processed in mammalian cells [11,12]. SUMO2 and SUMO3 share >95% sequence identity and often are referred to as SUMO2/3 collectively; however, they widely differ from SUMO1, with only 50% sequence identity [12,13]. The ubiquitin machinery contains dozens of E2s, but only one SUMO E2, ubiquitin-conjugating enzyme 9 (UBC9 encoded by *UBE2I*), which functions in eukaryotic cells [14]. The SUMO pathway achieves substrate specificity by regulating subcellular localization of a handful of E3 enzymes [15], whereas the ubiquitin system employs hundreds of E3s that provide selectivity [16]. Notably, modification with ubiquitin often results in proteasomal degradation. In contrast, attaching SUMO regulates protein localization, binding, and activity [14]. Ubiquitin chains come with different linkages (K6, K11, K27, K29, K33, K48, and K63), resulting in either linear or branched chains, and numerous examples describe how particular linkages determine the proteolytic or non-proteolytic fate of the target protein [17]. In contrast, one internal SUMO consensus site (ΨKxE, Ψ is large hydrophobic residue, x is any residue) at K11 in SUMO2/3 is utilized to build SUMO chains, whereas SUMO1 has an inverted internal consensus motif (ExK) [12,13,18]. Lastly, ubiquitination is normally highly regulated, and is induced by DNA damage or cell cycle activities, whereas SUMOylation and deSUMOylation are maintained at equilibrium under normal physiological conditions. Upon genotoxic stress, SUMO attachments are rapidly stabilized, leading to the activation of stress responses [19,20,21,22].

STUbLs are a unique class of E3 ubiquitin ligases because their action enables crosstalk between the SUMO and ubiquitin systems. *Saccharomyces cerevisiae* synthetic lethality of unknown [X] function 5 and 8 (Slx5 and Slx8) were identified as proteins of unknown function in a synthetic lethality screen with slow growth suppressor 1 (Sgs1), the homolog of mammalian Bloom syndrome protein (BLM) [23]. Subsequently, it was uncovered that *slx5* and *slx8* mutants accumulated SUMOylated proteins, suggesting that these two proteins are regulators of the SUMO pathway [24]. Interestingly, the depletion of two novel really interesting new gene (RING) finger proteins 1 and 2 (Rfp1 and Rfp2) in *Schizosaccharomyces pombe* showed a similar phenotype that could be complemented with *Homo sapiens* RING finger protein 4 (RNF4) [25]. These observations demonstrated that budding yeast Slx5/Slx8, fission yeast Rfp1/Rfp2/Slx8, and human RNF4 are evolutionarily conserved STUbLs [26,27,28]. Other STUbLs include ubiquitin ligase for SUMO conjugates (Uls1) and RING-type E3 ubiquitin transferase Rad18 (Rad18) in *S. cerevisiae* [29,30,31], degringolade (Dgrn) in *Drosophila melanogaster* [32,33], and RNF111/Arkadia in *H. sapiens* [34]. Here, we review the various processes in which these STUbLs have been implicated.

## 2. Structural Insights into STUbL Proteins and Regulation of STUbL Activity

An STUbL is characterized by one or multiple SUMO-interacting motifs (SIMs) that recognize SUMOylated substrates and a RING-type E3 ubiquitin ligase domain to catalyze the transfer of ubiquitin from the E2 conjugating enzyme to a lysine residue in the substrate [35,36,37] (Figure 1a). To date, the structures of the two human STUbLs have been resolved. In addition to SIMs and RING domains, RNF4 and RNF111 also have unique structural feature(s), namely the arginine-rich motif (ARM) [38] and arginine–lysine–lysine (RKK) [39] motif in RNF4 and the middle (M) region domain in RNF111 [40]. Below, we will review the structures and functions of the SIM, RING, ARM, RKK, and M domains in human STUbLs.

### 2.1. Architecture of SIMs

SIMs are typically composed of a hydrophobic core rich in valine, isoleucine, and leucine and are flanked by acidic residues that can potentially increase the affinity for SUMO via electrostatic interaction [47,48]. SIMs are intrinsically disordered in solution [47,49] but adopt a β-strand that binds to the β-sheet in the hydrophobic groove on SUMO [50]. SIMs interact with SUMO through medium to low affinity binding at micromolar concentrations [51,52,53,54]. The N-terminus of RNF4 contains four SIMs, presumably increasing affinity for poly-SUMOylated substrates. Indeed, the affinity for SUMOylated substrates is higher for full-length RNF4 than for any individual SIM deletion mutant, as determined by isothermal titration calorimetry (ITC) [54]. Among the four SIMs in RNF4, SIM2 and SIM3 have the highest affinity for SUMO chains and are the key determinants for SUMO chain binding and subsequent substrate ubiquitination. Mutations in SIM2 and SIM3 significantly lower the affinity for SUMO substrate [12,14,17] and abolish the in vitro [47] and in vivo [54] ubiquitination activity of RNF4. Furthermore, RNF4 preferentially binds to tetra-SUMO over di- or mono-SUMO as shown by ITC and nuclear magnetic resonance (NMR) titration [47,51,54]. These data support the model of multivalency of the SUMO–SIM interaction which results in a high avidity binding. Interestingly, RNF4 can also bind to a substrate that is mono-SUMOylated on multiple lysines, and SIM2 and SIM3 are still the most critical SIMs in this interaction [55].

In addition to disordered SIMs, the crystal structures of the tetra-SUMO2 (PDB code 1WM3) [51] and the di-SUMO2 chains (PDB code 4BKG) [54] indicate that SUMO chains are flexible and unrestrained. However, the solution NMR structure of di-SUMO2 in complex with RNF4-SIM2,3 (PDB code 2MP2) shows that both SUMO moieties bind to SIMs simultaneously and become oriented and restrained in the di-SUMO-RNF4 SIM complex [47]. Furthermore, binding to poly-SUMO chains induces the dimerization of RNF4, as measured by fluorescence resonance energy transfer, and subsequently activates its E3 ubiquitin ligase activity both in vitro and in vivo [56]. Although the N-terminal SIM cluster of RNF4 is structurally disordered, it adopts a compact global architecture to facilitate the transfer of ubiquitin from the C-terminal RING domain to substrates bound to the N-terminal SIMs [49]. Moreover, the inherent flexibility of the N-terminal SIM domains allows RNF4 to initiate ubiquitination on multiple substrates and grow long ubiquitin chains processively, leading to rapid amplification of the SUMO-ubiquitin signal [49].

RNF111 was identified as another human STUbL using a computational search for proteins that contain a SIM cluster similar to RNF4 [34]. RNF111 contains two SIMs and one “SUMO one binding” (SOB) motif [57] (Figure 1a), which specifically interacts with SUMO1 [58]. SUMO1, unlike SUMO2 and SUMO3, has an inverted internal consensus motif and does not form chains efficiently [59]. Therefore, SUMO1 often acts as a chain terminator when poly-SUMO chains are formed [60]. Due to its SOB motif, RNF111 has a strong preference for substrates bearing SUMO1-capped SUMO2/3 chains over pure SUMO2/3 chains both in vitro and in vivo, whereas RNF4 does not have this preference [57]. These studies suggest that the different SUMO binding motifs affect the specificity toward SUMOylated substrates.

### 2.2. Structural Characteristics of the RING Domain

There are two main types of E3 ubiquitin ligases: the homology to E6AP carboxyl terminus (HECT) and the RING E3 ligases [16,61]. The HECT E3 utilizes a two-step mechanism involving the formation of a transient thioester with the HECT domain. In contrast, the RING-type E3 catalyzes the direct transfer of ubiquitin from the E2 to the substrate [61]. Therefore, the RING E3 is an adaptor protein that brings ubiquitin-loaded E2 and a substrate into close proximity to catalyze the transfer. RING domains adopt a unique cross-brace structure and contain a conserved arrangement of cysteine and histidine residues coordinating two structural zinc (Zn^2+^) ions similar to Zn-fingers [61]. Sequence alignment suggests that the RING domains of STUbLs adopt the canonical 40–60 amino acid RING sequence, composed of seven conserved cysteine residues and one histidine (Figure 1b) [62]. RING E3 ligases can be monomeric or dimeric and function as single subunit E3s where binding to E2~ubiquitin and substrate occurs within the same polypeptide [50].

Both mouse (PDB code 3NG2) and rat (PDB code 2XEU) crystal structures of the RNF4 RING domain suggest that RNF4 acts as a dimer. Monomeric RNF4 mutants with a disrupted dimerization interface are inactive in an in vitro ubiquitination assay [63,64]. Furthermore, the RNF4 dimer preferentially binds to ubiquitin-loaded E2 over free E2 [29]. The crystal structure of the dimeric RING domain of rat RNF4 in complex with the ubiquitin-conjugating enzyme E2 D1 (UBE2D1/UbcH5A) linked to ubiquitin (RNF4 RING-UbcH5A~ubiquitin, PDB code 4AP4) shows that the E2 interacts with a single RNF4 protomer; however, two ubiquitin molecules bind to both protomers at the dimer interface (Figure 1c) [44]. This structure demonstrates why dimerization of the RNF4 RING is required for ubiquitination, a process that is induced by RNF4 binding to SUMO chains [56]. Ubiquitin conjugated to E2 retains flexibility and can adopt multiple conformations in the absence of an E3 enzyme [65]. The binding of RNF4 to UbcH5A~ubiquitin activates the thioester bond between E2 and ubiquitin by locking the C-terminal tail of ubiquitin into the active site of the E2 enzyme and attaching Gly76 to the substrate. This positioning optimizes the arrangement for a nucleophilic attack of ubiquitin by the incoming substrate lysine [44,64]. E2s that work with various STUbL proteins are summarized in Table 1. In conclusion, these studies support a model in which the RING-type E3 ligases play a more active role in catalyzing the transfer of ubiquitin from E2 to the substrate, rather than simply acting as a scaffold.

In contrast to RNF4, the solution structure of RNF111 (PDB code 2KIZ) suggests that its RING domain does not form a dimer, and size exclusion also supports the idea that native RNF111 exists as a monomer [45,75]. Interestingly, the crystal structure of the RNF111 RING domain in complex with the ubiquitin-conjugating enzyme E2 D2 (UBE2D2/UbcH5B) linked to ubiquitin (Ark2C-UbcH5B~ubiquitin, PDB code 5D0K) has revealed that in addition to the canonical E2~ubiquitin binding interface with the RING domain, the backside of the domain directly binds to a second ubiquitin (Figure 1d). This second ubiquitin also directly interacts with the ubiquitin moiety in the E2~ubiquitin complex to stabilize the closed conformation and promote ubiquitin transfer by RNF111. The mechanism utilized by RNF111 ensures the rapid formation of poly-ubiquitin chains onto mono-ubiquitinated substrates [45]. Additionally, a tryptophan (W972) in an α-helix in the C-terminus of RNF111 is essential for E2 recruitment and binding [75], and is conserved in monomeric RING-type E3 ligases [78]. The solution structure of a mutant RNF111 RING (W972R) (PDB code 5LG7) displays a distortion in the α-helix, disrupting its E2 binding activity. This mutation also significantly abrogates the function of RNF111 in mouse fibroblasts [76]. Interestingly, introducing a tryptophan into the corresponding α-helix in RNF4 (RNF4 S166W) significantly increases the ubiquitination activity and E2 binding affinity of RNF4. Furthermore, S166W bypasses the necessity of dimerization for the ubiquitination activity of RNF4 [78]. These studies suggest that the orientation of the α-helix in the C-terminus is the key factor that distinguishes monomeric from dimeric RING E3 ubiquitin ligases.

RNF4 can synthesize K11, K48, and K63-linked ubiquitin chains in vitro [46,64,79]. K63-linked ubiquitin chains are generated when it cooperates with both the E2 ubiquitin-conjugating enzyme E2 N (UBE2N/Ubc13) and ubiquitin-conjugating enzyme E2 variant 2 (UBE2V2/Mms2), a pseudo E2. The crystal structure of Ubc13~ubiquitin-RNF4 RING-Mms2 has been solved (PDB code 5AIT), and it has a very similar conformation to the crystal structure of UbcH5A~ubiquitin-RNF4-RING (Figure 1e) [44,46]. RNF4 also activates the thioester bond between the E2 Ubc13 and the donor ubiquitin. Furthermore, Mms2 brings a second (priming) ubiquitin and activates Ubc13 by inducing a catalytic configuration of the active site. The RING domain of RNF4 juxtaposes the K63 linkage of the priming ubiquitin and the donor ubiquitin in the E2 active site, allowing for rapid formation of K63-linked ubiquitin chains on a mono-ubiquitinated substrate [46], similar to RNF111 [45]. In summary, the wealth of structural information on human STUbL RING domains contributes to our understanding of the universal mechanism underlying RING E3 catalysis.

### 2.3. Unique Structural Motifs in Human STUbLs

RNF4 possesses an evolutionarily conserved, arginine-rich, highly positively charged ARM (residues 73–82), immediately following the fourth SIM [38]. Interestingly, the ARM is vital for RNF4 to recognize substrates that are simultaneously SUMOylated and phosphorylated, as the positively charged ARM interacts with negatively charged phosphorylated substrates. The ARM and SIMs of RNF4 cooperatively bind to substrates using a bimodular recognition mechanism that enhances substrate specificity [38]. Furthermore, the basic ARM surrounded by acidic residues promotes a compact conformation of the disordered N-terminus that allows RNF4 to efficiently transfer ubiquitin from the C-terminal RING to substrates bound to the N-terminal SIMs [49]. Thus, the ARM is not only important for substrate recognition but plays a critical role in maintaining the global architecture of RNF4.

The crystal structure of the human RNF4 RING domain has been solved (PDB code 4PPE). Both sequence analysis and the superimposition of the human RNF4 RING domain with two other RING-containing proteins, RNF2 and RNF168, have revealed a conserved, three basic residue cluster adjacent to the Zn^2+^-binding site of the RING domain [39]. The three basic RKK residues of RNF4 (residues 177–179) are predicted to bind DNA through electrostatic interactions. Indeed, the ability of RKK to bind DNA is required for RNF4 to ubiquitinate histone H3 within assembled nucleosomes in vitro and for the activity of RNF4 in the DNA damage response (DDR) in vivo [39]. Similarly, the twenty basic residue cluster located in the N-terminus of RNF4 is indispensable for in vitro DNA binding activity [80,81].

Human RNF111 is a much larger protein (994 amino acids) than human RNF4 (190 amino acids) and is characterized by the presence of a longer linker between the N-terminal SIMs and the C-terminal RING. The linker region (residues 415–865) has been designated as the M domain [40]. The M domain and SIMs of RNF111 contribute to both substrate recognition and subcellular localization. Furthermore, sequence analysis has revealed that the M domain is completely void of lysine but is significantly enriched for histidine and proline residues (poly-His motif) [40]. The poly-His motif promotes disorder and has been associated with the assembly of nuclear structures that repress gene expression [82,83]. Importantly, the presence of unique domains for RNF4 (ARM and RKK motifs) and RNF111 (M domain) suggests nonredundant roles for human STUbLs.

## 3. The Diverse Roles of STUbLs in Genome Maintenance

STUbL proteins play diverse roles throughout the cell cycle to protect against genome instability. This section details the function of STUbLs in DNA repair, DNA replication, mitosis, nuclear localization, promyelocytic leukemia protein (PML) homeostasis, and transcriptional regulation. STUbLs often function in specific environments, such as at centromeres or kinetochores, the nuclear periphery, and in PML nuclear bodies (PML-NBs). This compartmentalization of functions is linked to the role of SUMOylation and is vital for the repair of specific DNA lesions.

### 3.1. Functions in DNA Double-Strand Break (DSB) Repair

DSBs are one of the most toxic DNA lesions because they can result in chromosomal loss or rearrangements. Exogenous sources of DSBs include ionizing radiation (IR), high doses of ultraviolet (UV) light, and chemical agents, including various cancer chemotherapeutics, such as bleomycin and etoposide [84,85,86]. In addition, endogenous sources such as reactive oxygen species and replication stress resulting in fork collapse are major contributors [84,86]. It is estimated that a human cell experiences approximately ten DSBs per day; thus, DSB repair mechanisms are vital for maintaining genome stability [84,86]. Two major pathways repair DSBs, non-homologous end joining (NHEJ) and homologous recombination (HR). NHEJ is the main repair pathway during G_1_-phase in the absence of a sister-chromatid, whereas HR predominates during S- and G_2_-phases of the cell cycle when a repair template is available [87]. NHEJ ligates broken ends directly to each other, preventing aberrant chromosome translocations. The Ku70/80 heterodimer binds and protects DNA ends, but also provides a docking site for downstream repair proteins, such as DNA-dependent protein kinase catalytic subunit (DNA-PKcs), X-ray repair cross-complementing protein 4 (XRCC4), XRCC4-like factor (XLF/Cernunnos), and DNA ligase IV [84,85,86]. DNA-PKcs activity initiates a signaling cascade that phosphorylates members of the NHEJ machinery while p53-binding protein 1 (53BP1) blocks resection [84,85,86]. End processing is minimal but can lead to small insertions or deletions at the break site, rendering this pathway error-prone.

HR requires a homologous template for error-free repair. The meiotic recombination 11 homolog 1 (MRE11), ATP-binding cassette-ATPase (RAD50), and phosphopeptide-binding Nijmegen breakage syndrome protein 1 (NBS1) (MRN) complex senses DNA breaks and activates the kinase, ataxia telangiectasia mutated (ATM). ATM orchestrates the recruitment of several factors to the break site, including breast cancer type 1 susceptibility protein (BRCA1) to counteract 53BP1′s block of end resection [85,88,89]. Once end resection occurs, a cell is committed to HR. Initially, this is achieved by the MRN endonuclease in conjunction with CtBP-interacting protein (CtIP), but other nucleases catalyze extensive resection generating 3′ single-stranded DNA (ssDNA) [85,88,89]. Replication protein A (RPA) coats this ssDNA and recruits ataxia telangiectasia and Rad3-related protein (ATR) to ultimately promote cell cycle arrest [85,88,89]. Radiation sensitive 51 (RAD51) is loaded onto DNA by BRCA2 and/or RAD52, replacing RPA and generating RAD51 nucleofilaments [85,88,89]. These nucleofilaments invade the homologous sister chromatid, creating a displacement loop (D-loop) with a free 3′ end for extension of DNA synthesis [85,88,89]. Upon second end capture, a double Holliday junction is formed, which can be resolved to generate crossover or non-crossover products [85,88,89]. Several studies have elucidated the mechanisms underlying pathway choice. It is widely accepted that cyclin-dependent kinase (CDK) activity in S-phase drives end resection and commitment to HR [90,91,92,93].

#### 3.1.1. STUbLs in NHEJ

Although the budding yeast STUbLs, Uls1 and Slx5/Slx8, show structural similarities, they regulate NHEJ differently. Uls1 plays a specific role at telomeres where it inhibits NHEJ with the help of Ras-related protein 1 (Rap1) to avoid chromosome end-to-end fusions [94,95]. Rap1 SUMOylation at K240 and K246 cripples its ability to inhibit NHEJ [95]. SUMOylation recruits Uls1, which uses its translocase activity to detach SUMOylated Rap1 molecules from DNA and its E3 ubiquitin ligase activity to target Rap1 for proteasomal degradation (Table 2) [95]. This process allows unmodified Rap1 to bind DNA and prevent erroneous NHEJ [95]. Uls1 also has a specific role in downregulating “faulty” NHEJ at DSBs with telomeric repeat sequences. Upon inserting TG-rich repeats on one side of an inducible DSB, the broken DNA relocates to the nuclear envelope and requires Uls1 translocase activity to prevent aberrant NHEJ [96]. Importantly, this Uls1-dependent relocation does not occur at normal DSBs, but exclusively at breaks within telomeric repeats [96]. Slx5/Slx8 has been implicated in chromosome relocalization in the context of expanded CAG repeats that transiently tether to the nuclear pore, which requires nucleoporin 84 (Nup84) (further discussed in Section 3.4.3) [97].

Although few substrates have been identified, mammalian RNF4 plays an important role during NHEJ. Numerous studies have demonstrated that RNF4 depletion results in IR sensitivity and impaired NHEJ in several human cell lines [98,99,100,101,102]. Overexpressing sentrin-specific protease 2 (SENP2), a SUMO protease, resulted in a 2.5-fold increase in NHEJ [103]. It is possible that enhanced SENP2 activity prevents rapid accumulation of a SUMOylated substrate to avoid its early eviction from DSBs, but eventually requires RNF4-mediated removal for subsequent repair steps. Indeed, RNF4 depletion has a mild impact on XRCC4 recruitment to DNA, which can be restored upon depletion of ataxin-3, a deubiquitylating enzyme (DUB) that antagonizes RNF4 activity [104]. Thus, SENPs, DUBs, and STUbLs all reside at DSBs to provide temporal regulation of the NHEJ pathway.

Multiple independent studies confirm that RNF4 depletion delays 53BP1 recruitment and permits its persistence at DSB sites [39,98,101,105]. This is best described in a report that characterized a point mutation in RNF4, K179D, which abolishes the interaction between RNF4′s RING domain and DNA [39]. This mutant retains full E3 ubiquitin ligase activity but is unable to perform nucleosome-directed catalytic activity identifying a nucleosome targeting region in RNF4 [39]. In telomeric repeat-binding factor 2 (TRF2) conditional knockout MEFs, a model for telomere deprotection, RNF4 is required for 53BP1 recruitment, which results in telomere fusions [39]. Surprisingly, this also depends on RNF4′s SIM domains, reminiscent of Uls1′s role at telomeres [39]. However, 53BP1 has yet to be established as a direct RNF4 substrate (Table 2). Lastly, the lesser studied mammalian STUbL, RNF111, indirectly promotes NHEJ by participating in protein neddylation, which negatively regulates CtIP-mediated DNA end resection [106]. Overall, these studies highlight the importance of STUbLs in facilitating NHEJ but leave unanswered questions about their precise substrates and recognition factors in this pathway of DSB repair.

#### 3.1.2. STUbLs in HR

One of the first hints that STUbLs played a role in HR came from a study in fission yeast that identified Rfp1 and Rfp2 as checkpoint kinase 1 (Chk1)-interacting proteins [25]. Similar to HR mutants, rfp1Δ rfp2Δ double mutants showed poor growth, cell cycle delay, and sensitivity to IR and methyl methanesulfonate (MMS) [25]. These phenotypes are caused by an accumulation of SUMOylated proteins and can be rescued upon introduction of human RNF4 [25]. Recently, several groups have uncovered a role for Uls1, after D-loop formation in the removal of Rad51 to allow gap filling by DNA polymerases. This function is typically performed by Rad54 and Rdh54 proteins, but Uls1 substitutes in their absence to ensure that Rad51 is cleared from chromatin. These data suggest that Rad51 is a Uls1 substrate (Table 2) [110]. Two proteins that disentangle HR intermediates include Sgs1, the homolog of mammalian BLM, and mutagen-sensitive 81 (Mus81). Intriguingly, the lethality of sgs1Δ mus81Δ double mutants is rescued by inactivating Rad51 or Uls1 [111]. Therefore, deleting ULS1 partially impairs HR. In addition, other groups provide evidence that Uls1 participates in Holliday junction resolution. In fact, Uls1 physically interacts with the Holliday junction resolvase Yen1 [168].

Whereas Uls1 is necessary for proper HR, Slx5/Slx8 negatively regulates recombination. Upon deletion of *slx5/slx8*, spontaneous Rad52 foci increase accordingly with recombination rates [97,115]. Slx5/Slx8 targets SUMOylated Rad52 for degradation to allow fork restart and prevents illegitimate recombination upon DSB formation (Table 2) [97,115]. Consistent with this function, Slx5 co-localizes with Rad52 at DSBs [169]. Additionally, Slx5/Slx8 negatively regulates Sgs1, although Sgs1 is not a direct Slx5/Slx8 target [170]. Instead, it operates through Rad52 to control formation of Sgs1 foci, presumably to prevent improper recombination [170]. This function is echoed in the fly STUbL, Dgrn, which associates with DSBs at which SUMOylation blocks HR until Dgrn mediates their relocalization to the nuclear periphery to promote removal of SUMOylated proteins [171]. This HR obstruction and relocalization is consistent with Slx5/Slx8′s role in relocating expanded CAG repeats to the nuclear pore (discussed in Section 3.4.3) [97]. Importantly, this SUMO-induced HR block has been reported by other studies and spatially segregates DNA repair events to avoid untimely Rad52- or Rad51-mediated repair [172,173]. Therefore, both Slx5/Slx8 and Dgrn protect genome stability by relocating DSBs to the nuclear periphery for proper HR in a SUMO-depleted environment.

RNF4 has been implicated in regulating HR through the timely removal of mediator of DNA damage checkpoint protein 1 (MDC1) [105]. Indeed, MDC1 SUMOylated at K1840 is responsible for RNF4 recruitment to DSBs and is a direct substrate of RNF4 (Table 2) [98,100,101,105]. At DSBs, SENP2 and ataxin-3 closely monitor the timing of RNF4 activity and prevent premature removal of MDC1 by tightly regulating its SUMOylation and ubiquitination [103,104]. The recruitment of ataxin-3 not only depends on SUMOylation but also on a transient DNA damage-induced poly(ADP-ribosyl)ation (PARylation) signal [174]. The transient nature of the PARylation may provide an acute time window for ataxin-3 activity early in the DDR, whereas persistent SUMOylation allows RNF4-mediated turnover of MDC1, facilitating downstream repair steps [174]. Collectively, a picture is emerging in which multiple PTMs grant temporal and spatial precision for appropriate DSB repair.

Other studies have deepened our understanding of the function of RNF4 in the regulation of Kruppel-associated box (KRAB)-associated protein 1 (KAP1) and CtIP. RNF4 uses bimodular recognition of KAP1 phosphorylated at S824 (p-S824-KAP1), a DSB marker, by its ARM domain in combination with capturing SUMOylated KAP1 at K676 by its SIM domain [38]. Notably, p-S824-KAP1 is important to block HR and enable NHEJ in G_1_ [38]. As cells progress to S/G_2_-phase of the cell cycle, RNF4 accumulates, allowing RNF4-mediated and valosin-containing protein (VCP) segregase-assisted p-S824-KAP1 degradation (Table 2) [102]. This action relieves the KAP1-dependent brake on HR [102]. Similarly, RNF4 recognizes phosphorylated CtIP after protein inhibitor of activated STAT 4 (PIAS4) mediates its SUMOylation at K578 [131]. RNF4 then targets CtIP for degradation to limit excessive resection and ensure proper HR (Table 2) [131]. Interestingly, inhibiting ATM activity, which prevents CtIP phosphorylation and subsequent SUMOylation, blocks the interaction between CtIP and RNF4 [131].

Several groups have described roles for RNF4 in the recruitment of BRCA1 and turnover of RPA1. At DSB sites, RNF4 targets SUMO chains to create hybrid SUMO-ubiquitin tags recognized by receptor-associated protein 80, which is critical for the recruitment of BRCA1 [175]. It has been suggested that SUMOylated BRCA1 is an RNF4 substrate and its binding partner BRCA1-associated RING domain protein 1 (BARD1) is indirectly regulated by RNF4 through the degradation of PIAS1, a SUMO ligase (Table 2) [101,133]. Furthermore, RNF4 contributes to RPA1 turnover on ssDNA, which affects downstream Rad51 loading (Table 2) [98,105]. Since cells depleted for RNF4 fail to replace RPA with Rad51 on resected DNA, HR efficiency is compromised. Moreover, CDK2-mediated phosphorylation of T26 and T112 of RNF4 enhances its E3 ubiquitin ligase activity in S-phase [99]. Collectively, RNF4 is vital for proper HR repair by regulating MDC1, KAP1, CtIP, BRCA1, and RPA1. CDK2-dependent phosphorylation amplifies RNF4′s activity, providing a mechanism by which HR is bolstered in S-phase [99,102].

Lastly, RNF111 facilitates HR through regulating histone H4 neddylation at DSB sites. This damage-induced neddylation signal is mediated by RNF111 and recognized by the motif interacting with ubiquitin (MIU) domain of RNF168 [77]. Ultimately, protein neddylation is responsible for targeting or retaining a portion of RNF168 at DSBs [77]. Therefore, RNF111 assists in the signaling cascade which promotes HR.

#### 3.1.3. STUbLs in Other DNA Repair Pathways

Human RNF4 has been implicated in the removal of topoisomerase DNA–protein crosslinks (TOP-DPCs). The SUMO ligase PIAS4 is required for SUMOylation of both TOP1-DPC, trapped by camptothecin (CPT), and TOP2-DPC, trapped by etoposide [116]. This SUMOylation signal recruits RNF4, resulting in ubiquitination and subsequent degradation of both TOP1- and TOP2-DPCs (Table 2) [116]. Knocking down RNF111 has no effect on the ubiquitination status of TOP-DPCs, indicating this role is specific to RNF4 [116]. The SUMOylation of TOP-DPCs is not affected by exposure to inhibitors of replication, transcription, DNA-PKcs, ATM, or ATR [116]. Together, this implies that the SUMOylation of TOP-DPCs is independent of collisions with replication or transcription complexes and does not depend on activation of the DDR kinases [116]. It is plausible that this constitutes a repair mechanism designed to operate throughout the cell cycle, independently of the DDR, to ensure genome integrity [116]. It should be noted that deleting RNF4 does not fully block TOP-DPC removal, suggesting that other repair mechanisms exist [116]. Similar mechanisms exist in both budding and fission yeasts and act separately from the tyrosyl-DNA phosphodiesterase-dependent pathway for TOP-DPC repair [107,116,176].

In addition, RNF111 assists in nucleotide excision repair (NER). UBC13-MMS2 operates with RNF111 to form K63-linked ubiquitin chains on SUMOylated xeroderma pigmentosum, complementation group C (XPC) following UV damage [74]. RNF111 helps to recruit XPC to UV-induced DNA damage to facilitate NER [74].

### 3.2. Roles in DNA Replication and the Replication Stress Response

Accurate DNA replication is not only necessary for genome stability, but it is crucial for the reproduction of cells. This intricate process occurs during S-phase and involves three main steps: initiation, elongation, and termination [177,178,179,180]. Prior to initiation, replication origins must be properly “licensed” during late M- and G_1_-phases of the cell cycle, which involves loading the core of the replicative helicase, minichromosome maintenance protein complex 2-7 (MCM2-7) onto chromatin [177,178,179,180]. As cells transition from G_1_- to S-phase, two types of kinases, CDK and Dbf4-dependent kinase (DDK), act to recruit “firing” factors needed for initiation [177,178,179,180]. MCM10 is one essential firing factor and promotes DNA unwinding to allow for DNA synthesis to begin in a bidirectional manner [181,182]. DNA polymerase epsilon performs continuous leading strand synthesis, whereas DNA polymerase delta completes discontinuous lagging strand synthesis [181,182,183]. The sliding clamp, proliferating cell nuclear antigen (PCNA), enhances polymerase processivity and TOP1/2 relieves torsional strain so that elongation proceeds efficiently [177,178,179,180]. Lastly, replication termination ensues upon the convergence of two replication forks where ubiquitination of the MCM7 subunit promotes replisome disassembly mediated by the VCP segregase [179]. Together, these proteins safeguard the genome by producing a complete copy of the genetic material.

Repetitive sequences, DNA lesions, secondary DNA structures, and protein–DNA complexes can block the replication machinery, causing the slowing or stalling of replication fork progression [177,178,184,185]. This phenomenon is known as replication stress and elicits a replication stress response [177,178,184,185]. The uncoupling of the stalled DNA polymerase from the replicative helicase generates long stretches of ssDNA that are quickly coated by RPA [177,178,184,185]. This recruits and activates ATR which phosphorylates its targets, including CHK1, resulting in checkpoint activation, a decrease in origin firing, and stabilization of stalled replication forks [177,178,184,185]. Concomitantly, RPA recruits the E2/E3 complex, RAD6/RAD18, which mono-ubiquitinates PCNA, triggering DNA damage tolerance (DDT) pathways [177,178,184,185]. Mono-ubiquitination of PCNA promotes error-prone translesion synthesis (TLS), bypassing the lesion, whereas poly-ubiquitination of PCNA permits error-free template switching (TS) to repair the damage [177,178,184,185]. These mechanisms may involve replication fork reversal or remodeling by DNA helicases or translocases which promotes the restart of DNA synthesis [184]. Alternatively, prolonged fork stalling can result in fork collapse, either through fork instability or fork cleavage, generating a DSB which requires HR mechanisms for fork restart [178,184,185]. Therefore, the replication stress response governs a complex network that maintains genome stability.

#### 3.2.1. Yeast STUbLs That Operate in DNA Replication

The fission yeast STUbLs, Rfp1 and Rfp2, functional homologs of budding yeast Slx5, lack E3 activity but they recruit Slx8, through a RING–RING domain interaction, to form a functional E3 ubiquitin ligase [108,186]. *rfp1*Δ *rfp2*Δ double mutants display a severe growth defect and are exquisitely sensitive to hydroxyurea (HU) [108,186]. Consistently, fission yeast *slx8*Δ mutants grow slowly and are sensitive to UV light, HU, MMS, and CPT [107,108]. These phenotypes are suppressed upon deletion of the SUMO ligase Pli1, indicating that they are manifested through toxic accumulation of SUMOylated proteins [107,108]. One such substrate is Rad60, which averts the formation of toxic recombination-dependent structures upon fork stalling [108]. Slx8 ubiquitinates Rad60 in vitro in an Rfp1 SIM-dependent manner (Table 2) [108]. Furthermore, the interaction is evolutionarily conserved, as RNF4 co-precipitates NFAT-interacting protein (NIP45), the human homolog of Rad60, and binding is dependent on NIP45′s SUMO-like domains (Table 2) [108]. Intriguingly, Slx8, Rad60, and the fission yeast SUMO E3 ligase Nse2 cooperate to suppress TOP1-dependent DNA damage [176]. Consistently, Slx8 removes TOP1–SUMO conjugates at a replication fork block (RFB) to limit recombination (Table 2) [107]. Interestingly, the fission yeast SUMO-targeted DNA translocase Rrp2, homolog of Uls1, antagonizes Slx8-dependent TOP2 degradation [187]. By outcompeting Slx8 for SUMO-binding, Rrp2 displaces SUMOylated TOP2 from DNA through its translocase activity [187]. Thus, Rrp2 prevents exposure of TOP2-concealed DSBs which occur upon Slx8-mediated TOP2 degradation [187].

The role of Uls1 in the replication stress response is obscured by the fact that stalled replication forks can be converted to DSBs, requiring HR mechanisms for fork restart. Several groups have demonstrated that *ULS1* deletion rescues *sgs1*Δ mutant sensitivity to HU, which depends on Uls1′s translocase, SIM, and RING domains [111,188,189,190]. Interestingly, this rescue was not observed when *ULS1* was deleted from *top3*Δ and *rmi1*Δ mutants, indicating that Uls1 operates with Sgs1 outside of its role in Holliday junction resolution by the Sgs1-TOP3-RmiI (STR) complex [188]. Sgs1 promotes replication fork progression by non-recombinogenic mechanisms to prevent HR, as demonstrated in *sgs1*Δ mutants that exhibit a hyperrecombination phenotype at the ribosomal DNA (rDNA) locus, a region in the genome rich in natural RFBs [76]. Moreover, cells lacking both Uls1 and the endonuclease Mus81 are more sensitive to HU, MMS, and CPT than either single mutant. It has been proposed that Uls1 works with Sgs1 in a pathway that is parallel to Mus81 to resolve stalled replication fork intermediates [111,188,189,190].

The anti-recombinase, suppressor of Rad6 2 (Srs2), is a putative Uls1 target. In addition to ubiquitination, PCNA can be modified by SUMO, which suppresses HR during normal replication to favor DDT pathway activation at stalled forks [188]. This HR suppression is mediated by Srs2 binding to SUMOylated PCNA to disrupt Rad51 nucleofilaments [188]. The first hint that Uls1 played a role in this process was the observation that *uls1*Δ mutants have elevated levels of mono- and poly-ubiquitinated PCNA when compared to wild-type cells [188]. This suggests that *uls1* deletion may channel damage towards DDT pathways, resulting in the previously described *uls1*Δ *sgs1*Δ mutant resistance to HU. In line with this observation, Rad18 and Rad5 ubiquitin ligases, but not the TLS polymerase Rev3, are required to confer HU resistance [189]. Intriguingly, Srs2 is also modified by SUMO, which inhibits its interaction with SUMOylated PCNA [111]. Therefore, Uls1-mediated degradation of SUMOylated Srs2 promotes unmodified Srs2 binding to PCNA to suppress HR and promote TS, which enhances resistance to HU (Table 2) [111]. This recombination-based TS mechanism is especially important when Sgs1 or Mus81 function is compromised [111,190].

Unlike Uls1, deletion of which rescues *sgs1*Δ mutant phenotypes, inactivating Slx5/Slx8 is synthetically lethal with *sgs1*Δ [23]. Therefore, these two STUbLs are non-redundant and *uls1*Δ *slx5*Δ double mutants accumulate more SUMO conjugates than either single mutant [29,191,192]. Slx5/Slx8 participates in the replication stress response, as it effectively protects cells against the effects of chronic HU exposure and regulates sporulation efficiency and growth [23]. Interestingly, *slx5*Δ *slx8*Δ mutants are not sensitive to transient HU treatment, but require prolonged exposure to exhibit a significant effect on viability [193]. This suggests that Slx5/Slx8 plays a limited role in stabilizing stalled replication forks, but the complex is engaged upon fork collapse, which is consistent with data observed for RNF4 (discussed in Section 3.2.2) [193]. The *slx5*Δ *slx8*Δ mutants display a dramatic increase in spontaneous Ddc2 (ortholog of human ATR-interacting protein; ATRIP) foci and Rad53 phosphorylation [193]. This implies that deletion of *SLX5/SLX8* results in an increase in spontaneous DNA damage, which could explain the synthetic lethality observed with Sgs1 [193].

In addition, *slx5* and *slx8* are synthetically sick with *mcm10-1*, a temperature-sensitive allele of the essential initiation factor Mcm10 that causes under-replication [181,182]. The *mcm10-1 slx5*Δ double mutants have a significant growth defect characterized by a prolonged G_2_/M arrest, suppression of which requires a functional Slx5 RING domain [22]. Mechanistically, Slx5/Slx8 rescues viability of *mcm10-1* mutants by targeting the homolog of mammalian SURVIVIN, baculoviral IAP repeat-containing protein 1 (Bir1), and the mammalian Inner Centromere Protein (INCENP) homolog, synthetically lethal with *ipl1* (Sli15), for degradation (Table 2) [22]. Bir1 and Sli15 are both components of the chromosome passenger complex (CPC). Modest under-replication in *mcm10-1* cells at semi-permissive conditions fails to elicit an S/G_2_-phase arrest and cells enter mitosis with incompletely replicated chromosomes [22]. This triggers the spindle assembly checkpoint (SAC); however, cells evade mitotic arrest in the presence of Slx5/Slx8 (further discussed in Section 3.3.1) [22]. Importantly, this replication stress tolerance pathway is not specific to *mcm10-1* mutants, as the same phenotypes arise following MMS treatment [22]. Recently, our group has confirmed this negative genetic interaction between *MCM10* and *RNF4* in transformed mammalian cells (Oram and Bielinsky, unpublished).

During unperturbed DNA replication, Slx5/Slx8 regulates the turnover of SUMOylated DDK (Table 2) [117]. Interestingly, DDK is not SUMOylated in response to replication stress, but rather during normal replication to engage chromatin-bound DDK in origin firing [117]. SUMOylation of DDK is regulated by the SUMO protease Ulp2, which resides at replication origins to avoid unscheduled Slx5/Slx8-mediated degradation [117]. Presumably, SUMOylated DDK needs to be degraded to prevent re-replication and facilitate replication termination [117]. A proteomics screen suggests that several subunits of the MCM2-7 helicase may also be potential substrates of Ulp2 and Slx5/Slx8 regulation [117]. Perhaps this Ulp2-Slx5/Slx8 axis operates during replication termination as a backup pathway to Dia2-mediated Mcm7 ubiquitination [117]. This hypothesis requires further investigation, but would be consistent with the synthetic lethality between *dia2* and *slx5* mutants [194].

Budding yeast Rad18 is considered an STUbL, but this feature is not conserved in humans. Curiously, the SUMO E2, Ubc9, physically interacts with Rad18, which prompted the search and discovery of a Rad18 SIM domain [195]. Rad18 is well known for being the E3 ubiquitin ligase that ubiquitinates PCNA to trigger DDT pathways [31]. In budding yeast, SAP and msx-interacting zinc (mIZ)-finger domain 1 (Siz1) SUMO E3 ligase is responsible for attaching SUMO to PCNA at K164, the same residue that is mono- or poly-ubiquitinated by Rad18 [195]. SUMOylated PCNA recruits the anti-recombinase, Srs2, to prevent HR at stalled replication forks [195]. The interaction between Rad18-SIM and SUMOylated PCNA stimulates Rad18′s ubiquitin ligase activity (Table 2) [31]. Removing Rad18′s SIM domain produces a phenotype similar to DDT-deficient *rev3*Δ or *rad30*Δ mutants, implying that the SIM domain maximizes Rad18 activity [31]. As mentioned above, this SIM-dependent enhancement of Rad18 function is not conserved in humans, as mammalian Rad18 lacks a SIM domain [31].

#### 3.2.2. Functions of Mammalian STUbLs in DNA Replication

RNF4 plays a role in replication fork collapse, particularly when checkpoint function is compromised [196]. ATR conditional knockout MEFs treated with aphidicolin (APH) exhibit impaired replication fork restart, consistent with ATR’s known role in preventing replication fork collapse [196]. Intriguingly, RNF4 suppression restores the ability to synthesize DNA following fork stalling in ATR-deleted cells, shown globally by 5-ethynyl-2′-deoxyuridine (EdU) incorporation and at individual replication forks by DNA combing [196]. Importantly, reinitiation of DNA synthesis after removal of APH can be traced to reactivated replication forks and is not due to dormant origin firing. These findings suggest that RNF4 promotes replication fork collapse [196]. Although RNF4 suppression allows replication fork restart, these forks lack long-range processivity and only promote a brief surge in DNA synthesis [196]. One interpretation is that this restart is “illegitimate” since it cannot support prolonged DNA synthesis and results in reduced replication fidelity. Additionally, it suggests that RNF4 plays a role in DNA replication after restart [196]. Many components of the replisome are SUMOylated in response to replication stress, which could serve as a signal for RNF4-mediated turnover [196]. Presumably, degrading replisome components would either provide DNA access to structure-specific nucleases, or actively disassemble the replication machinery [196]. Although this role has been established at stalled/collapsed replication forks, it is possible that a similar mechanism exists at normal termination sites. RNF4 may behave similarly to what has been proposed for Slx5/Slx8, acting as a backup pathway for Mcm7 ubiquitination, but this has yet to be demonstrated.

RNF4 has also been connected to the Fanconi anemia (FA) pathway that is activated in response to interstrand crosslinks (ICLs). The FA pathway utilizes multiple repair mechanisms to perform DNA incisions, allow for lesion bypass, and complete lesion repair, by the consecutive action of NER, TLS, and HR proteins [197]. ICLs cause replication fork arrest and subsequent ATR activation to facilitate recruitment of the FA core complex, which consists of 14 FA proteins that function as a large ubiquitin ligase for FANCD2 and FANCI [197]. Mono-ubiquitination of FANCD2/FANCI locks the dimer onto DNA and allows for the recruitment of NER nucleases to mediate incisions proximal to the ICL [197]. The FANCD2/FANCI complex is SUMOylated in response to replication fork stalling by the SUMO E3 ligases PIAS1 and PIAS4. This reaction is antagonized by the SUMO protease SENP6 [134]. This tight SUMO regulation facilitates optimal timing of RNF4-mediated and VCP segregase-assisted degradation of the FANCD2/FANCI dimer, limiting its DNA damage-induced chromatin association (Table 2) [134]. Furthermore, RNF4 regulates the turnover of two FA core complex proteins, FANCA and FANCE (Table 2) [135]. RNF4 suppression increases sensitivity to the crosslinking agent mitomycin C and enhances chromosome aberrations and radials, all hallmarks of a clinical FA phenotype [135]. The degradation of these FA core complex components is likely needed for replication resumption following fork stalling, arguing that RNF4 is an important component of the FA pathway [135].

RNF111 plays a role in TLS in response to UV-light-induced DNA damage. DNA polymerase eta (pol η), a low-fidelity TLS polymerase, has high affinity for mono-ubiquitinated PCNA, but its access must be tightly controlled to prevent excessive mutagenesis [157]. Upon its association with mono-ubiquitinated PCNA, pol η is SUMOylated by PIAS1, promoting RNF111-mediated turnover after replication of damaged DNA (Table 2) [157]. Interestingly, knockdown of RNF4 or RNF111 leads to an increase in pol η foci, expression, and SUMOylation [157]. This effect by RNF4 is independent of its STUbL activity, whereas it requires the SIM and RING domains of RNF111 [157]. This implies that pol η is a direct target of RNF111, but not RNF4 [157]. Collectively, the mammalian STUbLs, RNF4, and RNF111 limit mutagenesis during TLS to safeguard the genome.

### 3.3. STUbL Participation in Mitotic Cell Division

Following chromosome duplication, cells must appropriately divide the genetic material into two identical daughters. Each pair of fully replicated sister chromatids is held together by cohesion complexes [198]. At mitotic onset, chromatin is highly condensed, and kinetochores assemble on the centromeres of each sister chromatid, providing an attachment site for microtubules to the spindle poles [198,199,200,201,202]. Following proper spindle attachment, chromosomes migrate to the spindle equator and align along the metaphase plate [198,202]. Sister chromatids begin to separate as they are pulled to opposite ends of the cell during anaphase [198,199,201]. Lastly, a new nuclear envelope develops around each set of separated daughter chromosomes and cytokinesis “pinches off” the separated nuclei forming two identical daughter cells [198,200].

Aberrant mitosis can result in chromosome missegregation, manifesting as micronuclei or aneuploidy, which are both characteristics of cancer cells [199,200,201,203,204]. Two sophisticated mitotic machineries, the correction mechanism and the SAC, certify the proper connection of microtubules to kinetochores and delay cell division until accurate chromosome segregation can be guaranteed [198,200,201,203,204]. By destabilizing microtubules at kinetochores, the correction mechanism disrupts incorrect attachments, keeping the SAC active until bi-orientation of sister kinetochores to opposite spindle poles is achieved [198,199,201]. This process is thought to occur by a “search and capture” mechanism mediated by the CPC, which resides at the centromere–kinetochore interface [198,199,202,203,204]. The SAC inhibits anaphase onset by preventing the ubiquitin ligase activity of the anaphase-promoting complex (APC/C) through targeting its coactivator Cdc20 [198,201,203,204]. Once bi-orientation is satisfied, the SAC is turned off, allowing Cdc20 to activate APC/C, which degrades cyclin B1 and securin [198,199,201,203,204]. Cyclin B1 degradation inactivates CDK1, inducing late mitotic events, and securin degradation activates the protease separase, which cleaves cohesion proteins, allowing sister chromatid segregation [198,201,203,204]. Protein phosphatase 2A (PP2A) supports this metaphase-to-anaphase transition by dephosphorylating Cdc20, which promotes its interaction with APC/C [204]. In summary, the correction mechanism and SAC preserve genome stability by making sure each daughter cell receives the correct number of chromosomes.

#### 3.3.1. STUbLs in Kinetochore- or Centromere-Specific Processes

Several studies have revealed that STUbLs reside and function at kinetochores and centromeres, highlighting their role in chromosome stability. Genome-wide binding analysis in budding yeast shows that Slx5 is enriched at centromeres, exclusively centered on the core centromere but not the pericentromeric region [205]. A yeast strain defective in kinetochore assembly abolishes Slx5 centromeric binding, implying that this unique localization of Slx5 is kinetochore-dependent [205]. Both *slx5*Δ and *slx8*Δ mutants display extreme mitotic defects, including aneuploidy, spindle mispositioning, fishhooks, and abnormal spindle kinetics [205]. Intriguingly, both *slx5* and *slx8* strains display a positive genetic interaction with *rts1*, a regulatory subunit of PP2A phosphatase [205]. Indeed, Slx5 and Slx8 mutants accumulate Rts1 at centromeres during metaphase and display variable distance between sister chromatids, potentially experiencing a loss of centromeric tension [205]. In fact, inactivating Slx5 prevents the degradation of SUMOylated Mcd1, a cohesin subunit, and causes an accumulation of condensin subunits (Table 2) [118,206]. Furthermore, the budding yeast homolog of securin, Pds1, is stabilized in *mcm10-1 slx5*Δ mutants, confirming that cohesion cleavage is perturbed and responsible for prolonged mitotic arrest [22]. Additionally, Slx5/Slx8 degrades CPC components, Bir1 and Sli15; thus, several STUbL substrates reside at the kinetochore, coinciding with Slx5 localization (Table 2) [22,205].

Slx5/Slx8 regulates the function of the centromere-specific histone H3-like protein Cse4 (CENP-A in humans, Table 2) [119,120,121,122,207]. Mislocalization of Cse4/CENP-A is linked to aneuploidy in yeast and tumorigenesis in humans [119]. Cse4 is modified on K65 by the SUMO ligases Siz1 and Siz2, which induces degradation by Slx5/Slx8 [119,120,121]. Importantly, proteolysis of Cse4 prevents its mislocalization to euchromatin [119]. In parallel to Slx5/Slx8, the E3 ubiquitin ligases Ubr1, Psh1, and Rcy1 promote Cse4 turnover, underscoring the importance of properly regulating this substrate [120,121]. Intriguingly, two other lysine residues of Cse4, K215 and K216, are also SUMOylated but do not promote Slx5/Slx8 or other E3-mediated turnover [122]. Rather, the modification serves to facilitate its genome-wide chromatin deposition [122]. Whether or not RNF4 targets CENP-A for degradation in humans remains unknown but is an important question to address.

Mammalian RNF4 has been shown to control kinetochore function, ensuring that this complex is properly assembled at the appropriate time and location. Cells lacking SENP6, a SUMO protease, display defects in spindle assembly and metaphase chromosome alignment [136]. Further inspection of kinetochore composition shows that these cells lack any detectable signal for the inner kinetochore proteins, CENP-H/I/K [136]. Typically, the CENP-H/I/K complex localizes to centromeres during S-phase, which coincides with the appearance of SUMOylated CENP-I [136]. Indeed, RNF4 degrades CENP-I, a process that is antagonized by the activity of SENP6 (Table 2) [136]. Chromosome misalignment increases in RNF4- or SENP6-depleted cells and markedly decreases upon their co-depletion, confirming that these proteins act antagonistically [136]. Ultimately, this mechanism balances kinetochore assembly for faithful chromosome segregation.

#### 3.3.2. STUbLs in Chromosome Segregation

In fission yeast, *rfp1*Δ *rfp2*Δ double mutants display fragmented chromosomes, elongated nuclei, and asymmetric positioning of nuclei [186]. *S. pombe slx8* mutants exhibit missegregated chromosomes and multinucleated cells, which is suppressed by deleting the SUMO ligase Pli1 [107]. The importance of STUbLs in mitosis appears to be evolutionarily conserved as knockdown of RNF4 leads to an increase in chromosome bridges resulting in segregation errors [205]. One potential substrate important in this context is the histone variant H2A.Z that has been reported to physically interact with STUbLs and is a target of SUMOylation (Table 2) [112]. Histone H2A.Z is essential for the accurate transmission of chromosomes and accumulates in G_2_/M-phase cells in three deletion strains, *uls1*Δ, *slx5*Δ, and *slx8*Δ cells [112,113]. Overexpression of histone H2A.Z causes a significant mitotic delay, indicating that it must be controlled for normal cell cycle progression [112]. A mass spectrometry method that identified proteins co-modified by SUMO and ubiquitin revealed the mitotic regulators KIF23 and MIS18BP1 as potential RNF4 targets (Table 2) [137]. KIF23 belongs to the family of microtubule-dependent molecular motors, and MIS18BP1 is required for normal chromosome segregation [137]. Knockdown of RNF4 leads to an accumulation of KIF23 and MIS18BP1 in mitosis and aberrant chromosome partition [137]. Therefore, STUbL proteins regulate various mitotic substrates to ensure proper chromosome division.

Budding yeast Slx5/Slx8 plays a specific role in suppressing gross chromosomal rearrangements (GCRs). Deleting *SLX5* or *SLX8* dramatically increases GCRs 296- and 152-fold over wild-type cells, respectively [193,208]. In *slx5*Δ mutants, 75% of GCRs and in *slx8*Δ mutants, 53% were de novo telomere additions [193]. Moreover, methyl methanesulfonate sensitivity 21 (Mms21)-dependent SUMOylation that acts upstream of Slx5/Slx8 largely suppresses duplication-mediated GCRs [206]. Whether these GCRs arise from deficiencies in DSB repair, elevated replication stress, impaired cohesion cleavage, or other kinetochore-related substrates remains ambiguous. To further confound this, Slx5/Slx8 targets phosphorylated Mms4 (Table 2) [123]. The Mus81-Mms4 nuclease is activated in G_2_/M-phase by Mms4 phosphorylation to process replication-associated recombination structures and avoid chromosome segregation errors in mitosis [123]. Indeed, Mms4 undergoes SUMOylation and subsequent Slx5/Slx8-mediated turnover assisted by the Cdc48 segregase under unperturbed conditions [123]. If Mms4 is not degraded in mitosis, active Mus81-Mms4 nuclease persists into subsequent cell cycle stages, during which its aberrant activity could cause illegitimate DNA cleavage and rearrangements [123].

RNF4 promotes chromosome stability by preventing the loss of whole chromosomes. RNF4-null chicken DT40 cells exhibit substantial increases in chromosome breaks and, unexpectedly, DNA content dropped 25% at six weeks post-*RNF4* deletion [209]. Mitotic chromosome spreads show 60% of RNF4-null cells lost an entire copy of chromosome II [209]. These cells are very sensitive to colcemid, a drug that prevents microtubule polymerization, signifying a potential SAC defect [209]. Furthermore, RNF4-null cells display premature separation of sister chromatids which, together with an SAC defect, could result in the loss of whole chromosomes [209]. Treatment with the potent TOP2 inhibitor etoposide causes RNF4 to localize along the axis of mitotic chromosomes [210]. Remarkably, depleting SUMO2/3 abolishes RNF4 recruitment to etoposide-exposed mitotic chromosomes, suggesting that this localization is SIM-dependent [210]. Furthermore, RNF4-deficient cells often contain micronuclei, indicative of abnormal chromosome segregation and perturbed mitotic events [210]. Taken together, mammalian RNF4 participates in multiple mitotic processes to ensure accurate chromosome segregation.

### 3.4. STUbLs Localize Damaged DNA to the Nuclear Periphery

The nuclear periphery, composed of a double membrane nuclear envelope (NE) and multiple nuclear pore complexes (NPCs) [211], has emerged as a central player in the maintenance of genome stability [212,213]. Defects in nuclear peripheral components have been implicated in a diverse set of diseases such as cancer, premature aging, and neurological disorders [214,215]. The nuclear periphery not only provides spatiotemporal regulation of DNA replication timing [216], but also serves as a site for DNA damage repair. It was first shown in budding yeast that efficient repair of DSBs within subtelomeric DNA requires tethering of telomeres to the NPC, specifically the Nup84 complex. Failure to tether telomeres to the NPC by disrupting the Nup84 complex leads to decreased survival [217]. Several DNA lesions, including damaged rDNA, irreparable or heterochromatic DSBs, eroded telomeres, as well as collapsed and stalled replication forks, relocate to the nuclear periphery for proper repair. The relocalization of damaged DNA from the nucleoplasm to the nuclear periphery has been revealed in yeast, flies, and mammalian cells, suggesting that it is a conserved mechanism. Although the repair of different types of DNA damage at the nuclear periphery has distinctive requirements and mechanisms, STUbLs play an integral role in the relocalization process per se and in the regulation of the repair itself. Furthermore, Slx5/Slx8-mediated ubiquitination of nucleoporin 60 (Nup60) strengthens its association with Nup84 and thus facilitates the functions of NPCs in response to DNA damage [218]. Below, we review how STUbLs participate in repairing different types of DNA damage by modulating lesion relocalization (Figure 2).

#### 3.4.1. Processing of DSBs in rDNA, Heterochromatin, and Irreparable DSBs

The repair of DSBs in the highly repetitive yeast rDNA depends on controlled HR. Relocalization of damaged rDNA outside the nucleolus, where it normally resides, suppresses uncontrolled HR (Figure 2a) [226]. Similarly, repair of DSBs in heterochromatin in flies and MEFs by HR requires that DSBs move outside of the heterochromatin domain before Rad51-mediated strand invasion occurs, avoiding the use of a nonhomologous chromosome as a template. Interestingly, SUMOylation mediated by the Smc5/6 complex, a scaffold for Mms21, is needed for these processes in both yeast and flies (Figure 2b) [222,227,228]. In yeast, irreparable DSBs relocate from the nucleoplasm to the nuclear pores or the inner nuclear membrane protein (INMP), mono polar spindle 3 (Mps3), in an Slx5/Slx8-dependent manner (Figure 2c) [219,220,229]. The target destination for DSBs on the nuclear periphery is determined by the cell cycle and the SUMOylation status of proteins bound to broken DNA. Persistent DSBs in S-phase cells are mono-SUMOylated by Mms21 and relocate to Mps3, whereas breaks in G_1_-phase cells are modified with poly-SUMO chains by the sequential SUMOylation activity of Mms21 and Siz2. Slx5/Slx8 specifically recognizes poly-SUMOylated proteins and recruits associated DSBs to nuclear pores [221,230]. In flies, Dgrn is responsible for moving SUMOylated, heterochromatic breaks to both the nuclear pores and Mps3 (Koi and Spag4 in flies). Mms21 (Cervantes and Quijote in flies) and PIAS (dPIAS in flies) coordinate the SUMOylation of proteins at DSBs prior to Dgrn recruitment (Figure 2b) [171,231]. In addition to SUMOylation, another requirement for persistent DSBs to relocate is end processing. Once the breaks reach the nuclear periphery, STUbLs facilitate the degradation of SUMOylated proteins that block late HR steps, allowing for Rad51 loading and subsequent repair of DSBs primarily via break-induced replication (BIR) or microhomology-mediated end joining (MMEJ) (Figure 2c) [221]. Interestingly, in mouse cells, if heterochromatic breaks fail to localize to the nuclear periphery, they associate with the single-strand annealing (SSA) factor RAD52 and undergo mutagenic repair [232]. These studies demonstrate that the spatial separation between end resection and homology search is necessary to ensure high-fidelity repair of heterochromatic breaks.

#### 3.4.2. Processing of Eroded Telomeres

The clustering of budding yeast telomeres at the NE is dependent on the interaction between telomerase and Mps3. Disruption of the telomerase–Mps3 interaction enhances recombination, suggesting that telomere anchorage to Mps3 suppresses recombination during replication and maintains telomere stability [222]. When telomerase is absent, telomeres gradually erode and cells eventually enter irreversible growth arrest (i.e., senescence) [233]. Nonetheless, rare survivors escape growth arrest by recovering telomere length with recombination-based mechanisms [234]. Short telomeres first utilize type I Rad51-dependent recombination to extend. However, at the later stage of senescence, the efficiency of type I recombination diminishes and unresolved recombination intermediates accumulate at telomeres [224,235]. Telomeric proteins are SUMOylated upon erosion [223,236] and SUMOylation of these unrepairable telomeres is enhanced, triggering Slx5/Slx8-mediated relocation to the NPC for type II, Rad59-dependent recombination [224,236]. Type II recombination in yeast is similar to the mechanism utilized by telomerase-negative human cells that employ alternative lengthening of telomeres (ALT), a BIR-related process [236]. Slx5/Slx8 and Ulp1 SUMO protease at NPC potentially facilitate the removal of SUMOylated proteins to maximize type II recombination [224].

#### 3.4.3. Processing of Collapsed and Stalled Replication Forks

Collapsed or stalled replication forks caused by repetitive telomeric sequences, structure-forming sequences (i.e., expanded trinucleotide repeats), and RFBs localize to the nuclear periphery. In yeast, telomeric stalled forks relocate to the NPC for proper fork restart. Disruption of the NPC engages error-prone sister chromatid exchange (resulting from misalignment) to maintain minimal telomere length [237]. In human cells, telomere replication defects induced by mutations in protection of telomeres 1 (POT1) trigger ATR-dependent relocalization of dysfunctional telomeres to the NP. Depletion of an NPC subunit in these mutants leads to telomere fragility and elevated sister chromatid exchanges [238], demonstrating that NPC suppresses uncontrolled telomere recombination. Interestingly, RNF4 has a negative genetic interaction with *POT1* mutations [238], suggesting that this STUbL might also have a role in relocating dysfunctional telomeres to the nuclear periphery in human cells.

Expanded CAG repeats can lead to replication fork collapse and are relocated to the NPC in budding yeast. CAG repeats exhibit increased fragility and instability (expansions or contractions) in the absence of Nup84 or Slx5/Slx8 [97]. Collapsed forks at CAG repeats are resected by Mre11, Exo1, and Sgs1 to generate ssDNA, which is then bound by RPA, Rad52, and Rad59. These proteins are subsequently SUMOylated by Mms21. Mono-SUMOylation is sufficient for Slx5/Slx8 to facilitate the relocation of these collapsed forks, as *smt3-3KR* (SUMO that cannot form poly-SUMO chains) mutants do not exhibit relocalization defects [225]. It is therefore possible that Slx5/Slx8 recognizes multiple mono-SUMOylated proteins simultaneously or interacts with multiple mono-SUMO tags on a given protein. Following relocalization to the NPC, Rad52-SUMO is targeted by Slx5/Slx8 for proteasomal degradation. Rad51, which is initially blocked by RPA-SUMO, associates with CAG repeats at the NPC and promotes fork restart, possibly in concert with the Srs2 helicase (Figure 2e) [225]. Lastly, in contrast to expanded CAG repeats, arrested forks caused by RFBs in fission yeast require the loading of Rad51 and its strand invasion activity prior to being tethered to the NPC for recombination-dependent replication fork restart [172]. Arrested forks that cannot be rescued due to the failure of Rad51 loading are converted into sister chromatid bridges in mitosis [173]. Although Pli1-dependent poly-SUMO chains are required for anchoring RFBs to the NPC, they are toxic to recombination-mediated restart. Thus, Rfp1/Rfp2/Slx8 is required for both the relocalization of paused forks and the degradation of SUMOylated proteins at the NPC to facilitate repair [172,239]. SUMOylated proteins can also undergo direct SUMO removal by SENP protease Ulp1, which is constitutively attached and stabilized at NPCs by Nup132. Delocalization of Ulp1 by depleting Nup132 leads to a reduced frequency of fork restart. Interestingly, depletion of Pli1 can rescue restart defects seen in *nup132*Δ cells, suggesting the need to anchor to the NPCs is dispensable when poly-SUMOylation is absent [172]. Together, these results suggest that the anchorage of an active RFB to the NPC for restart through Pli1-dependent poly-SUMOylation is the preferred mechanism to resume replication and Slx5/Slx8 promotes genome integrity by fulfilling two main functions: (1) tethering arrested forks to the NPC and (2) removing SUMO chains that inhibit recombination-mediated restart (Figure 2f).

### 3.5. STUbLs Maintain the Homeostasis of Promyelocytic Leukemia Nuclear Bodies

PML is the first identified target of human RNF4 (Table 2) [79,108,138]. PML is a multifunctional protein and can polymerize into nuclear bodies called PML-NBs [240]. PML-NBs are dynamic, membraneless insoluble structures that host more than 150 proteins either transiently or permanently [241]. The nucleation of PML-NBs and recruitment of partner proteins are highly dependent on SUMO metabolism and multiple SUMO–SIM interactions between PML and its binding partners [242,243]. PML-NBs have been implicated in numerous cellular processes such as protein modification, transcriptional regulation, DNA damage response, and senescence [244,245]. PML-NBs serve as protein reservoirs to either redistribute or sequester proteins to regulate cell cycle functions or respond to stress. For example, mitotic checkpoint protein BUB3, a component of the SAC, is protected in PML-NBs from cullin 4-RING ubiquitin ligase (CRL4)- cell division cycle protein 2 (CDT2)-mediated degradation during interphase. This protection mechanism ensures that cells express sufficiently high levels of BUB3 before entering mitosis. At the onset of mitosis, BUB3 will dissociate from PML-NBs to activate the SAC [246]. PML degradation releases pro-apoptotic factors and induces apoptosis in response to arsenic treatment [219], a standard therapy for the treatment of acute promyelocytic leukemia (APL) [247]. Interestingly, arsenic-induced PML degradation is dependent on RNF4 [79,138]. The recruitment of RNF4 to PML-NBs relies on its SIM domains [248]. RNF111, the other human STUbL, also facilitates the degradation of PML upon arsenic exposure, independently of RNF4 (Table 2) [158]. RNF111 preferentially degrades PML modified with SUMO1-capped SUMO2/3 chains, in contrast to RNF4, which degrades PML modified with SUMO2/3 chains [57], suggesting that the SUMO composition regulates the activation of the RNF4 and RNF111 pathway.

PML possesses SUMO E3 ligase activity [249]. In response to oxidative stress, PML becomes oxidized and recruits UBC9, the universal SUMO E2 conjugating enzyme [139]. Other E3 SUMO ligases, such as PIAS1 [250], RAN binding protein 2 (RANBP2) [251], and zinc finger protein 451 (ZNF451) [252] are also associated with PML-NBs. PML is SUMOylated at three main lysines: K65, K160, and K490 [253]. To date, there are two proposed models of how arsenic treatment triggers RNF4-mediated degradation of SUMOylated PML. First, PML K65 is modified by SUMO2 during unstressed conditions [254,255]. Upon exposure to arsenic, PML K65-SUMO2 is replaced by SUMO1 by SUMO-specific peptidase 1 (SENP1). K65-SUMO1 promotes poly-SUMO2/3 chain formation at K160, triggering RNF4-mediated degradation. Therefore, K65 acts as a SUMO switch [254]. Second, PML K160 is first modified by a newly identified poly-SUMO5 chain but is then gradually replaced by poly-SUMO2/3 chains by SENP1 in response to arsenic. RNF4 preferentially interacts with SUMO2/3 but not SUMO5 and rebuilding the SUMO2/3 chain at K160 induces RNF4-mediated degradation of PML [255]. It is yet unclear whether these two mechanisms occur independently or collaboratively.

RNF4 promotes not only the degradation of PML, but also several PML-NB resident proteins, such as SP100, thymine DNA glycosylase (TDG), and homeodomain-interacting protein kinase 2 (HIPK2) (Table 2) [139]. The fact that STUbLs regulate the homeostasis of PML-NBs suggests that they can modulate a broad range of biological processes in which PML-NBs are involved [152,256,257,258]. Many DNA damage checkpoint and repair proteins are associated with PML-NBs, including DNA TOP II-binding protein 1 (TOPBP1), ATR, ATM, CHK2, p53, MRN, BRCA1, RAD51, RAD52, and BLM. Therefore, PML plays an important role in genome maintenance (reviewed in [245]). A recent study showed that either overexpression or depletion of PML impairs HR, suggesting that PML abundance has to be tightly controlled in response to DNA damage [259]. Quantitative SUMO proteomics following MMS treatment revealed that PML is indeed degraded in an RNF4-dependent manner [149]. Furthermore, PML-NBs are recruited to persistent DNA lesions via the RNF168-53BP1 pathway [260,261]. RNF168 specifically binds to the SUMO-ubiquitin hybrid chains generated by RNF4 [262]. Thus, RNF4 facilitates the association of PML-NBs with DNA damage sites. Lastly, in ALT cells, telomeres are clustered in ALT-associated PML bodies (APBs) [263]. APBs are required for recruiting the BLM-TOP3A-RMI1/2 (BTR) complex to chromosome ends in order to maintain telomere length and heterogeneity [264]. Telomeric DNA synthesis in APBs by BIR induces PIAS4-mediated SUMOylation of TRF2, further recruiting DDR proteins to APBs via SUMO–SIM interactions [265]. Given that Slx5/Slx8 promotes type II recombination at eroded telomeres in budding yeast, it will be worthwhile investigating whether human STUbLs also play roles in ALT telomere synthesis in APBs.

### 3.6. STUbLs Regulate Gene Transcription

Human RNF4 was initially identified as a transcriptional coregulator [69,266,267,268,269,270]. To date, many STUbLs, including Slx5/Slx8 in budding yeast, Dgrn in flies, and both human STUbLs regulate transcription by targeting a variety of factors involved in tissue development, the response to oxidative stress and heat shock, and tumorigenesis. One of the most frequently overexpressed proto-oncogenes, c-Myc [271], is targeted by RNF4 [140,272]. RNF4 either promotes c-Myc degradation (Table 2) [140] or stabilizes the protein by catalyzing an unusual K11/K33-linked poly-ubiquitin hybrid chain [272]. These findings suggest that the internal linkages of ubiquitin chains can determine whether RNF4 promotes protein degradation or stabilization (further discussed in Section 5) [272]. Other proto-oncogenes that are ubiquitinated by RNF4 include nuclear factor kappa B (NF-κB)-signaling pathway proteins [273] and cancer-associated specific protein 1 (Sp1) (Table 2) [141]. RNF4 suppresses canonical NF-κB signaling by targeting TAK1-TAK1-binding proteins 2 (TAB2) for lysosome-dependent degradation [273]. RNF111 acts as a positive regulator of the transforming growth factor β (TGF-β) signaling pathway by targeting the following repressors: small mothers against decapentaplegic 7 (SMAD7), c-Sloan-Kettering Institute (c-Ski), and Ski novel (SnoN) for degradation (Table 2) [40,161,164,166,274,275,276]. Notably, oncogene-induced replication is a significant source of replication stress that can ultimately lead to genome instability [4,277,278].

In response to hypoxic and heat shock stress, RNF4 targets hypoxia-inducible factor 2-alpha (HIF2-α) [143], factor inhibiting HIF 1 (FIH1) [144], and poly(ADP-ribose) polymerase 1 (PARP1) [142] (Table 2). PARP1 is not only a DDR protein, but also suppresses the expression of the heat shock protein 70.1 (HSP70.1). Upon heat shock, RNF4 targets SUMOylated PARP1 for degradation, and this leads to HSP70.1 gene activation. Thus, RNF4 functions as a positive regulator of HSP70.1 [142]. PARP1 has also been identified as a target of RNF4 in an independent proteomics study [133]. Nuclear factor erythroid 2-related factor 2 (NRF2), an essential transcription factor that functions in response to oxidative stress, is ubiquitinated by both RNF4 [145] and RNF111 [159] (Table 2). SUMOylated NRF2 shuttles into PML-NBs and is targeted by RNF4 for degradation. Therefore, RNF4 is a negative regulator of the transcription activity of NRF2 [145]. In contrast, RNF111 ubiquitinates and stabilizes SUMOylated NRF2 in the PML-NBs to enhance its transcription activity [159]. These studies suggest that although STUbLs share substrates, the consequences of their ubiquitination can have opposite outcomes for protein stability. How the balance of the ubiquitination catalyzed by the two STUbLs is maintained, has yet to be elucidated.

STUbL-regulated transcription is also important for the growth and development of flies [33] and life cycle of budding yeast [66,67,124,125,126,127]. During fly embryogenesis and embryonic segmentation, Dgrn targets Hairy and its co-repressor, Groucho [32], as well as enhancer of split complex E (spl)-C proteins, which are involved in Notch signaling [33] (Table 2). Modifier of transcription 1 (Mot1) is an essential protein [279] in budding yeast and associates with TATA-binding protein (TBP) to facilitate RNA polymerase II initiation [280]. Mot1 is SUMOylated by Siz1 and Siz2 and subsequently targeted by Slx5/Slx8 for degradation (Table 2). Interestingly, Slx5/Slx8 preferentially ubiquitinates the misfolded form of Mot1 over wild-type, suggesting a role of STUbLs in protein quality control [66]. A similar mechanism has been observed in humans where RNF4 and PML cooperatively degrade misfolded proteins [256,258]. Mating-type-specific gene alpha 2 (MATα2), a transcriptional repressor that regulates mating type switching in yeast, is targeted by Slx5/Slx8 [125,127] (Table 2). MATα2 contains a SUMO-like domain that is recognized by SIMs in Slx5/Slx8 [126], and SUMO modification of MATα2 is dispensable for Slx5/Slx8-dependent ubiquitination [67,126]. In contrast to MATα2, MATα1 can also be ubiquitinated by Slx5/Slx8 but in a SUMO-dependent manner [124] (Table 2).

## 4. Crosstalk between STUbLs and Other SUMO and Ubiquitin Metabolic Processes

SUMO and ubiquitin metabolic circuits play important roles in genome maintenance [7,8,9,281,282]. We and others have previously described the way that STUbLs work cooperatively with SUMO ligases, including PIAS family members (Siz1 and Siz2 in budding yeast, Pli1 in fission yeast, and dPIAS in flies) and NSMCE2 (Mms21 in budding yeast, Nse2 in fission yeast, and Cervantes and Quijote in flies). Interestingly, STUbLs directly regulate the SUMOylation machinery in yeast and humans [109,130,133]. The budding yeast Siz1 E3 SUMO ligase can be ubiquitinated by Slx5/Slx8 in vitro and in vivo. The interaction between Siz1 and Slx5/Slx8 is SUMO-dependent, and SUMOylated Siz1 accumulates in *slx5*Δ cells, suggesting that the auto-SUMOylation activity of Siz1 leads to its degradation through the STUbL-mediated pathway. Slx5/Slx8 ensures the timely degradation of Siz1 at the onset of mitosis, presumably to avoid the accumulation of nuclear SUMO conjugates that might interfere with cell cycle progression (Table 2) [130]. Similarly, fission yeast E3 SUMO ligase Pli1 has been identified as an STUbL target (Table 2) [109]. More recently, the human SUMO conjugation machinery, including the E2 conjugating enzyme UBC9 and five different E3 SUMO ligases (PIAS1, PIAS2, PIAS3, NSMCE2, and ZNF451), have been identified as substrates of RNF4 in a quantitative proteomics study (Table 2) [133]. Furthermore, RNF4 can indirectly affect the accumulation of SUMOylated BARD1 by regulating PIAS1 levels at DSB sites [133].

SENPs and STUbLs antagonize each other by competing for SUMOylated substrates. In yeast, Ulp2 protects SUMOylated DDK from Slx5/Slx8-mediated degradation until cells enter mitosis [117,283,284]. Similarly, Ulp2 stabilizes TOP I-interacting factor 2 (Tof2) at the nucleolus, a key factor ensuring rDNA silencing, whereas Slx5/Slx8 targets it for degradation (Table 2) [128,129]. In humans, the retention of MDC1 and ubiquitinated FANCD2/FANCI complex at damaged sites in DSB repair and ICL repair, respectively, are maintained through the balance between SENPs and RNF4 [103,134]. These studies highlight the importance of the balance between SUMOylation and deSUMOylation. Curiously, SENP6 has been implicated in maintaining a basal level of RPA70 SUMOylation under unperturbed conditions. However, in the presence of DNA damage, SENP6 dissociates from RPA70, allowing it to be SUMOylated and recruit RAD51 for HR [285]. Whether the recruitment of RAD51 by SUMOylated RPA is followed by the degradation of RPA mediated by RNF4 remains a subject of debate [98,285].

The human genome encodes approximately 100 DUBs [281], some of which are of particular interest here, as they antagonize STUbLs, as mentioned above. Both ubiquitin-specific peptidase 7 (USP7) and ubiquitin-specific peptidase 11 (USP11) hydrolyze the SUMO-ubiquitin hybrid chains that RNF4 generates in vitro [286,287]. Additionally, USP11 stabilizes PML-NBs in vivo and therefore counteracts RNF4-mediated degradation of PML [286,288]. Furthermore, RNF4-mediated removal of SUMOylated MDC1 from DSB sites is antagonized by USP7 and ataxin-3 [104,174,289]. The SUMO deubiquitinase (SDUB) activity of USP7 has rendered it an essential replisome-associated component for DNA replication by maintaining a SUMO-rich and ubiquitin-poor environment at active replisomes. Depletion of USP7 leads to slowed replication fork progression and reduced origin firing [287]. Additionally, USP7 has been implicated in numerous genome maintenance pathways, including cell cycle regulation, DDR, and telomere maintenance, as well as its well-characterized function in regulating p53 [290].

## 5. STUbLs Function in Protein Stabilization

Although seemingly contradictory to typical STUbL function, some STUbLs stabilize proteins rather than degrade them. RNF4 ubiquitination sustains otherwise transitory proto-oncogenic transcription factors, such as β-catenin, c-Myc, c-Jun, and the Notch intracellular domain (N-ICD) protein [272,291]. This activity requires substrate phosphorylation, rather than SUMOylation, which permits RNF4 binding through its ARM domain and stabilization via its E3 ubiquitin ligase activity [272,291]. Notably, these phosphorylations are catalyzed by mitogenic kinases, which display aberrant activity in tumorigenesis [291]. In line with its stabilizing effect, the ubiquitin linkages generated by RNF4 utilize K11 and K33, not K6, K27, K29, K48, or K63 [272,291]. RNF4 knockdown leads to substantial cell death in MDA-MB-231 breast cancer cells and prevents colony formation in soft agar [272]. Furthermore, high RNF4 messenger RNA and protein levels are found in luminal A malignant breast tumors and adenocarcinomas of the colon, respectively, and correlate with a decrease in overall survival [272]. These findings suggest that RNF4 promotes cancer cell survival by stabilizing and enhancing the activity of proto-oncogenes.

RNF4 promotes tumorigenesis and therapeutic resistance in melanoma cells, xenograft mouse models, and patient-derived cancer samples [292]. These phenotypes rely on the translation initiation factor, elongation factor alpha (eIF2α). RNF4 ubiquitination stabilizes eIF2α which requires substrate phosphorylation and interaction with RNF4′s ARM domain [292]. Consistently, patient-derived melanomas display elevated RNF4 and corresponding phosphorylated eIF2α levels, which correlate with poor prognosis and resistance to mitogen-activated protein kinase (MAPK) inhibitors [292]. RNF4-expressing tumors elevate the endothelial marker CD31 and vascular endothelial growth factor (VEGF), which promotes angiogenesis, a phenomenon that depends on RNF4′s RING activity [292]. This suggests that RNF4 may promote cancer progression, providing another example of a positive feedback mechanism for a proto-oncogene that is stabilized by RNF4 [292].

## 6. Concluding Remarks

STUbLs participate in diverse functions that help protect the integrity of the genome. They are highly specific towards their substrates and their actions are often balanced by SENPs and DUBs to regulate the half-life of SUMO conjugates. This provides the temporal and spatial control necessary for a dynamic chromatin environment during various steps of DNA metabolism. However, several gaps remain in our current understanding of STUbL regulation. For instance, RNF4 plays a role in NHEJ, but limited substrates have been identified [39,98,99,100,101,102,105]. Both budding yeast STUbLs, Uls1 and Slx5/Slx8, function specifically at telomeres, a region of the genome in which NHEJ must be suppressed to avoid chromosome fusion [94,95,96,224,236]. Indeed, RNF4 has been implicated in PML-NB homeostasis [79,108,138,248,254,255] and it is reasonable to hypothesize that it may participate in ALT telomere synthesis. Studies in budding yeast have provided insight into the relationship between STUbLs and Sgs1/BLM and will guide future investigations in mammalian cells. For example, Uls1 and Slx5/Slx8 show opposing genetic interactions with Sgs1, confirming that they function as non-redundant STUbLs [23,29,111,188,189,190,191,192]. This may also be true for mammalian RNF4 and RNF111. In human cells, RNF4 regulates the formation of BLM foci in PML-NB [170]; however, it is currently unclear whether RNF111 affects this process.

Other known STUbL functions also warrant further investigation. For example, RNF4 promotes replication fork collapse in ATR-deficient MEFs, but the precise replisome substrates of RNF4 at stalled forks have remained elusive [196]. Uncovering these targets would provide important mechanistic insights. Moreover, both yeast and human STUbLs prevent the loss of whole chromosomes and assist in chromosome segregation [107,112,123,137,186,193,205,208,209,210]. Whether these defects arise from insufficient DSB repair, enhanced replication stress, and/or altered cohesion cleavage will deepen our understanding of how STUbLs maintain genome stability. Lastly, both RNF4 and RNF111 stabilize proteins rather than degrade them depending on the specific ubiquitin linkages they catalyze. Deciphering how this process is regulated will provide new inroads into our understanding of cell transformation and tumorigenesis, as many of the known targets are proto-oncogenes [159,272,291,292]. Interestingly, overexpression of RNF4 is observed in lung, breast, and colon cancer cells [153,272]. Animal models for RNF111 argue that it can prevent or advance tumorigenesis depending on the specific context and cell type [293,294]. Outside of cancer, aberrant RNF4 function has been implicated in neurodegenerative disease caused by failure to protect against the accumulation of misfolded proteins [156] and in cardiac dysfunction [257,295]. Given the diverse roles for STUbLs in human physiology, a better understanding of their biology will inform whether this knowledge can be biomedically exploited in the future.

## Figures and Tables

**Figure 1 ijms-22-05391-f001:**
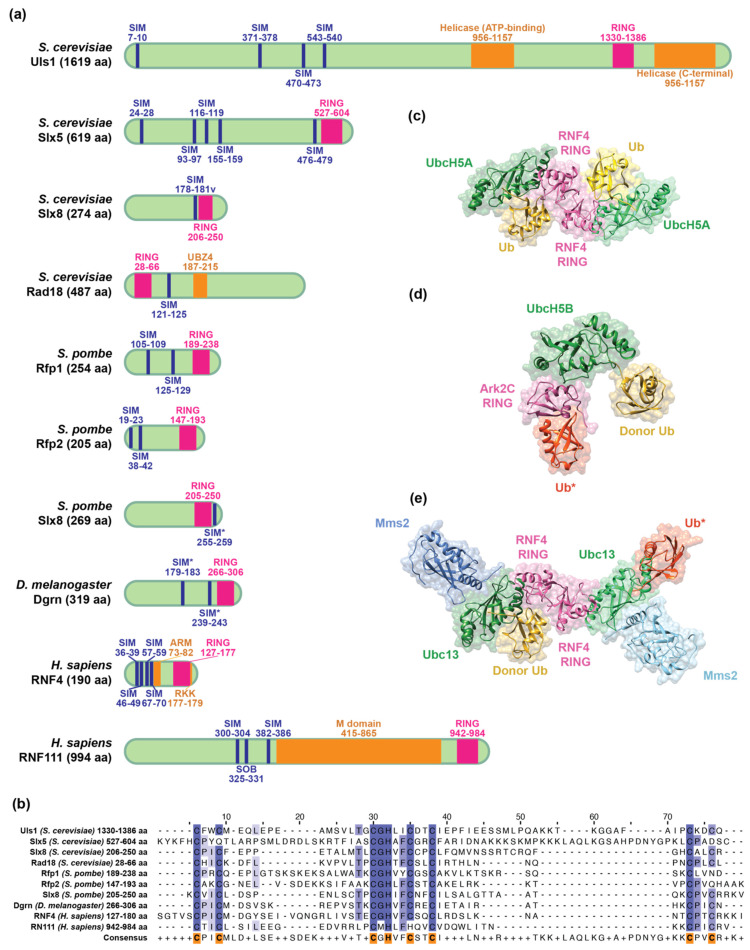
Overview of STUbLs. (**a**) The domain structures of STUbLs from *Saccharomyces cerevisiae*, *Saccharomyces pombe*, *Drosophila melanogaster*, and *Homo sapiens* are shown. Domain information was obtained from UniProt (https://www.uniprot.org/ accessed on 15 March 2021) or NCBI Protein (https://www.ncbi.nlm.nih.gov/protein/ accessed on 15 March 2021) databases. For STUbLs for which SIM information was unavailable, the predicted SIM sequences (SIM*) were obtained from GPS-SUMO [41,42]. (**b**) Sequence alignment of the RING domains of STUbLs. Amino acids are highlighted in blue based on their similarity. The seven conserved cysteine residues and one histidine residue in a canonical RING sequence are highlighted in orange in the consensus sequence. Sequence alignment was performed using Clustal Omega (https://www.ebi.ac.uk/Tools/msa/clustalo/ accessed on 15 March 2021) and Jalview [43]. (**c**) The crystal structure of RNF4 RING (pink)-UbcH5A (green)~ubiquitin (gold) (Protein Data Bank (PDB) code 4AP4 [44]). (**d**) The crystal structure of RNF111 RING (Ark2C, pink)-UbcH5B (green)~donor ubiquitin (gold) (PDB code 5D0K [45]). A second ubiquitin (Ub*, orange) that directly binds to the back side of RNF111 RING is shown. (**e**) The crystal structure of Ubc13 (green)~ubiquitin (gold)-RNF4 RING (pink)-Mms2 (blue) (PDB code 5AIT [46]). The priming ubiquitin (Ub*, orange) is shown. The structures were prepared with the Chimera program (http://www.cgl.ucsf.edu/chimera/ accessed on 15 March 2021). Abbreviations: SIM, SUMO-interacting motif; RING, really interesting new gene E3 ligase domain; aa, amino acids; UBZ4, ubiquitin-binding zinc finger type 4 domain; ARM, arginine-rich motif; RKK, arginine–lysine–lysine motif; SOB, SUMO one binding motif; M domain, middle domain.

**Figure 2 ijms-22-05391-f002:**
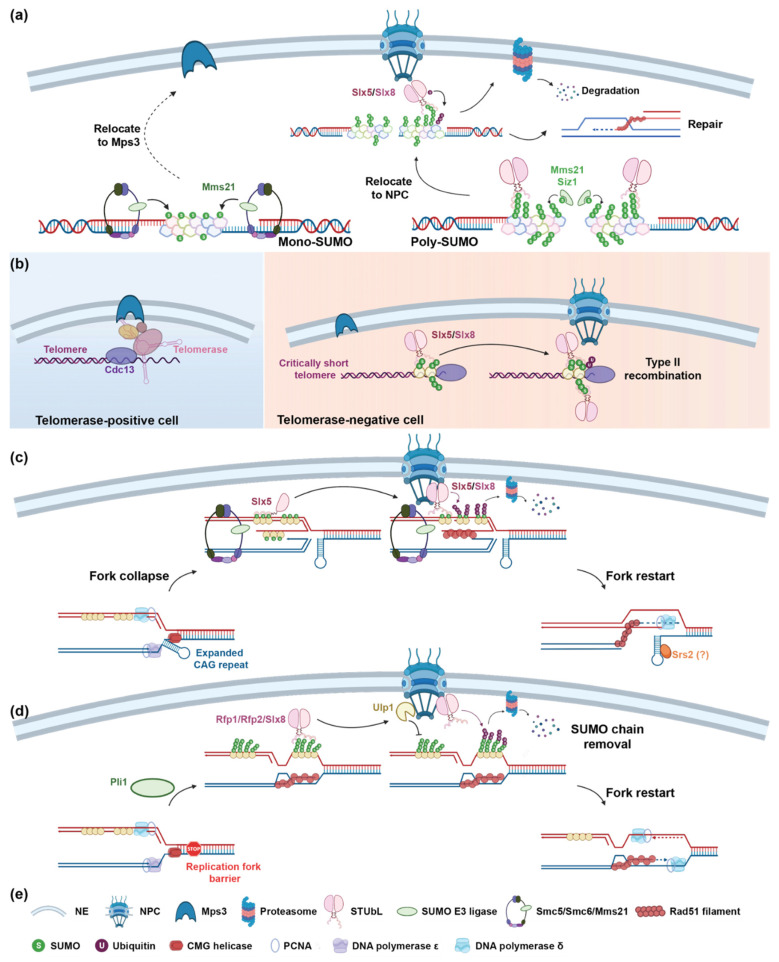
Relocalization of damaged DNA to the nuclear periphery by STUbLs. (**a**) Relocalization of irreparable DSBs to the nuclear periphery in *S. cerevisiae*. Irreparable DSBs that are mono-SUMOylated are relocated to Mps3, whereas DSBs that are poly-SUMOylated are relocated to the NPC in an Slx5/Slx8-dependent manner. Once DSBs are at the NPC, Slx5/Slx8 promotes the degradation of proteins blocking late HR steps, enabling the loading of Rad51 and subsequent repair [219,220,221]. (**b**) Relocalization of eroded telomeres to the nuclear periphery in *S. cerevisiae.* In telomerase-positive cells (left), telomeres are associated with NE via the interaction between telomerase and Mps3. In the absence of telomerase, Slx5/Slx8 recognizes SUMOylated proteins that bind to critically short telomeres and relocates them to NPC, at which type II recombination occurs to extend telomere lengths [222,223,224]. (**c**) Relocalization of collapsed forks to the NPCs in *S. cerevisiae*. Replication forks are prone to collapse at the expanded CAG repeats. Collapsed forks are processed and bound by RPA, Rad52, and Rad59, which are mono-SUMOylated by Smc5/Smc6 and Mms21. Collapsed forks relocate exclusively to the NPCs in an STUbL-dependent manner. Slx5/Slx8 also facilitates the degradation of Rad52 at the NPCs, allowing the association of Rad51 and subsequent fork restart, potentially with the assistance of Srs2 helicase [97,225]. (**d**) Relocalization of arrested forks to the nuclear periphery in *S. pombe*. Arrested forks caused by replication fork barriers are poly-SUMOylated by Pli1 SUMO E3 ligase. Rad51 loading and activity are required before relocalization by Rfp1/Rfp2/Slx8. At the NPC, both Ulp1 SUMO-specific peptidase and STUbL-proteasome can remove poly-SUMO chains, which are inhibitory to fork restart [172]. (**e**) Key to proteins and compartments shown in (**a**–**d**). Figures were created with BioRender.com. Abbreviations: STUbL, SUMO-targeted E3 ubiquitin ligase; Mps3, mono polar spindle 3; DSB, DNA double-strand break; NPC, nuclear pore complex; HR, homologous recombination; NE, nuclear envelope; RPA, replication protein A; Smc5/Smc6, structural maintenance of chromosome protein 5/6; Mms21, methyl methanesulfonate sensitivity 21; CMG, Cdc45-Mcm2-7-GINS; PCNA, proliferating cell nuclear antigen; Siz1, SAP and mIZ-finger domain 1; Srs2, suppressor of Rad6 2.

**Table 1 ijms-22-05391-t001:** Identified E2s that work with STUbL proteins either in vitro or in vivo (cell-based assay).

STUbL	Species	E2 (Ubiquitin Linkage If Known) [Reference]
Slx5/Slx8	*S. cerevisiae*	Ubc1 [36], Ubc4 [30,36,66,67], Ubc5 [36], Ubc13-Mms2 (K63) [36], UbcH6 [36], UbcH13-Uev1a (K63) [36]
Rad18	*S. cerevisiae*	Rad6 [31]
RNF4	*H. sapiens*	Ubc4 ^1^ (K63) [68], UbcH5a ^2^ [44,46,69,70,71], UbcH5b ^1^ [63,69,70], UbcH5c ^3^ [70], MmUbc7 ^4^ [69], Ubc13 ^5^ (K63) [46,69,72,73], Mms2 ^6^ (K63) [46], Ubc16 ^7^ [72,73], Rad6B ^8^ [69,70], E2-25K ^9^ [69]
RNF111	*H. sapiens*	UbcH5b ^1^ [74,75,76], UbcH5c ^3^ [74,75], Ubc13 ^5^-Mms2 ^6^ (K63) [74], Ubc12 ^10^ [77]

Alternative protein/gene names: ^1^ UBE2D2; ^2^ UBE2D1; ^3^ UBE2D3; ^4^ UBE2J2; ^5^ UBE2N; ^6^ UBE2V2; ^7^ UBE2W; ^8^ UBE2B; ^9^ UBE2K; ^10^ UBE2M.

**Table 2 ijms-22-05391-t002:** Substrates of STUbL proteins.

STUbL (*Species*)	Biological Process	Substrate (Tier ^1^) [Reference]
Rfp1/2, Slx8 (*S. pombe*)	DNA repair	TOP1 (2) [107]
	Replication	Rad60 (1) [108], TOP1 (2) [107]
	SUMO regulation	Pli1 (2) [109]
Uls1 (*S. cerevisiae*)	DNA repair	**HR:** Rad51 (3) [110]; **NHEJ:** Rap1 (2) [95]
	Replication	Srs2 (2) [111]
	Chromosome segregation	H2A.Z (3) [112,113]
Slx5/Slx8 (*S. cerevisiae*)	DNA repair	**HR:** Yen1 (1) [114], Rad52 (2) [97,115], RPA (1) [115], Rad59 (2) [115]; **Other repair:** TOP1-DPC (2) [116], TOP2-DPC (2) [116]
	Replication	Srs2 (2) [111], DDK (2) [117]
	Mitosis	**Kinetochore-specific:** Bir1 (2) [22], Sli15 (2) [22]; **Centromere-specific:** Mcd1 (2) [118], Cse4 (2) [119,120,121,122]; **Chromosome segregation:** H2A.Z (3) [112,113], Mms4 (2) [123]
	Transcription	Mot1 (2) [66], Matα1 (2) [124], Matα2 (1) [125,126,127], Tof2 (2) [128,129]
	SUMO regulation	Siz1 (1) [130]
Rad18 (*S. cerevisiae*)	Replication	PCNA (1) [31]
Dgrn (*D. melanogaster*)	Transcription	Hairy (1) [32], Groucho (2) [32], spl-C proteins (2) [33]
RNF4 (*H. sapiens*)	DNA repair	**HR:** MDC1 (1) [98,100,101,105], RPA1 (2) [98,105], BRCA1 (2) [101], KAP1 (2) [102], CtIP (1) [131]; **NHEJ:** TRF2 (2) [132], XRCC5 (3) [133], 53BP1 (3) [39]; **Other repair:** TOP1-DPC (1) [116], TOP2-DPC (1) [116]
	Replication	NIP45 (2) [108], FANCD2 (2)/FANCI (1) [134], FANCA (2) [135], FANCE (n/a) [135], BLM (3) [133], RAD18 (2) [133]
	Mitosis	**Kinetochore-specific:** CENP-I (2) [136]; **Chromosome segregation:** KIF23 (2) [137], MIS18BP1 (2) [137]
	PML homeostasis	PML (1) [79,108,138], SP100 (2) [139], TDG (2) [139], HIPK2 (2) [139]
	Transcription	c-Myc (2) [140], Sp1 (2) [141], PARP1 (2) [142], HIF-2α (2) [143], FIH1 (2) [144], NRF2 (2) [145], HNF4α (1) [146], ZNF746 (2) [147], MeCP2 (1) [148], KDM5B (2) [149,150], SETDB1(2) [133]
	SUMO/Ub regulation	**SUMO:** UBC9 (3) [133], PIAS1 (2) [133], PIAS2 (3) [133], PIAS3 (3) [133], NSMCE2 (3) [133], ZNF451 (3) [133]; **Ub:** USP7 (3) [133]
	Viral, cancer, and disease	**Viral:** Daxx (2) [151], Tax (1) [68,152]; **Cancer:** NDRG2 (2) [153], CBX2 (2) [154], PIM1 (2) [155]; **Disease:** ATXN7 (2) [156]
RNF111 (*H. sapiens*)	Replication	Pol η (2) [157]
	PML homeostasis	PML (2) [158]
	Transcription	NRF2 (2) [159], SMAD7 (2) [160,161,162], c-Ski (2) [160,163,164,165], SnoN (2) [160,163,164,166]
	Cancer	SMAD7 (2) [160,161,162], c-Ski (2) [160,163,164,165], SnoN (2) [160,163,164,166], ESRP2 (2) [167]

^1.^ Three tiers were used to describe each target: (1) substrates confirmed by in vitro ubiquitination assays, (2) substrates characterized using a cell-based assay by STUbL depletion/mutation, proteasome inhibition, cycloheximide chase assay, and/or co-immunoprecipitation experiments, and (3) uncharacterized substrates identified by proteomic screens or hypothesized by authors. Abbreviations: HR, homologous recombination; NHEJ: non-homologous end joining; PML, promyelocytic leukemia protein; Ub, ubiquitin; (n/a), data are not available (unpublished).

## Data Availability

Not applicable.

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
