# Peer review of "SUMO-Targeted Ubiquitin Ligases and Their Functions in Maintaining Genome Stability"

_ijms, 2021, doi:10.3390/ijms22105391_

Round 1

Reviewer 1 Report

The review article by Chang and Oram et al. entitled ‘SUMO-targeted ubiquitin ligases and their functions in maintaining genome stability’ gives an overview on the multifaceted aspects in which STUbLs are known to promote stable genome propagation in different organisms. This includes DNA repair and replication processes, mitosis, nuclear architecture, epigenetics etc. Overall, the topic is very timely and the article written in an engaging way. After a general introduction, which sets the scene nicely, the authors give a structure-function-related overview of STUbLs, which helps the reader follow the following topical sections on different genome stability aspects. Importantly, in addition to providing an in-depth summary of what is known in the field, the authors draw interesting and insightful conclusions on recent developments and elaborate on key questions that remain unanswered, which will be helpful for directing future research and appreciated by the community.

Below is a list of points for further consideration by the authors.

The Figures are beautiful and clear, conveying key points made in the article. Perhaps an additional Figure could help capture even more the diverse and complex genome stability processes covered in the article in a snapshot-type of way.

Line 27: Regarding DNA repair throughout the cell cycle: it would be beneficial to separate mitosis out from this; see e.g. here: ‘https://pubmed.ncbi.nlm.nih.gov/32001093/’.

Line 43: what about SUMO5? It comes up later in the article.

Line 45: not only polyubiquitin, but also fusions with ribosomal proteins; it would be good to either explain this more, or perhaps delete as it does not seem to play a major role afterwards.

Line 47: the different nomenclatures of E2 enzymes currently used in the literature are confusing – to facilitate ease of which enzymes are referred to, it would be beneficial to include the most commonly used names the first time they are mentioned e.g. UBC9, also known as UBE2I, or something along those lines (see also comment below).

Line 49: this paragraph misses out on referring to SUMO E3s to make a fair comparison between the Ub and SUMO conjugation systems in terms of numbers etc. Since SUMO E3s come up later in the article, it would further help set the scene regarding this early on.

Line 52: include linear chains.

Line 56: mention the absence/presence of consensus sites on the other SUMO isoforms already here to prepare the reader for what comes later?

Line 56: not sure exactly what ‘ubiquitination is limited’ means in this context; best to rephrase to clarify.

Figure 1 line 91: current predictors of SIMs such as GPS SUMO and JASSA are extremely unreliable and no fair comparison to validated SIMs. Distinguishing more the SIMs in the Figure would help the reader judge the reliability of the indicated SUMO-binding regions. For example, different colours could be used for validated versus GPS-SUMO-, JASSA- (or both GPS-SUMO-/JASSA-) predicted SIMs.

Line 102: abbreviation for RING is different here to the one used in the main text.

Line 107 etc.: best to be less specific regarding the SIM structure e.g. distinguishing I/V from L in most positions but one is misleading with regards to what is known about SIMs.

Line 113 etc.: perhaps bring in the term avidity in this context.

Line 175: it sounds as if the Gly76 is already attached when it is locked in – best to rephrase.

Line 181: it would be good to clarify what ‘in vivo’ refers to here (and elsewhere e.g. lines 197, 339), and perhaps even indicate for each E2 which one it applies to directly in the Table.

Table 1: see comment above regarding E2 nomenclature; it would helpful for readers accustomed more to one or the other nomenclature i.e. UBCx versus UBE2x; to extend the names to both nomenclatures. Perhaps a suitable place to do so would be in a footnote.

Table 1: ubiquitin linkage for Ubc13-Mms2 for RNF111 is missing.

Line 265: NHEJ -> abbreviation here is different to the one used in the Table 2 footnote; ‘non-homologous end joining’ is the more commonly used one.

Line 271: ‘DNA-dependent protein kinase catalytic subunit (DNA-PKcs)’ instead of ‘DNA-dependent protein kinase (DNA-PK)’ to distinguish between the catalytic subunit and the holocomplex, which is referred to as DNA-PK; needs to be adjusted afterwards accordingly, too.

Line 272: add XLF/Cernunnos as corresponding protein names for XRCC4-like factor.

Line 335: the precise questions that remain could be extended, as they are not limited to substrates, but also recognition factors etc.

Line 351: Sgs1 comes up in the beginning (line 64), too, where it might already be helpful to define its homologue.

Line 1003-1006: it is not clear what the additional roles are in this context; perhaps best to rephrase for clarity.

Author Response

Our response is uploaded in a word file.

Reviewer 2 Report

This review is by far the most comprehensive, scholarly and insightful review on STUbLs and their roles in genome stability and DNA repair. The authors literally summarized all the published work on STUbLs and discussed all aspects of STUbLs: structures, assembly and functions (291 references for a small family of ubiquitin ligase!). I also learned a lot while going through this manuscript. The tables (especially the substrates of STUbLs) are extremely thorough and informative; the figures are just elegant and illustrative.

Upon reading through the article, there were a few places in which minor changes could be made to clarify or improve the manuscript. Overall, the article is remarkably free of misspellings and grammatical errors:

  1. The authors may want to touch upon the similarities and differences among SUMO1, SUMO2, SUMO3 and SUMO4 in the introduction.
  2. Line 141: “…SUMO1, unlike SUMO2 and SUMO3, lacks the internal consensus SUMO modification site…”, SUMO1 has an inverted internal consensus motif EAKP that is responsible for SUMO1 polymerization (PMID: 23708367, PMID: 16428803).
  3. In section 3.1.2, it is worth mentioning that RNF111 acts as a NEDD8 E3 ligase that neddylates histone H4 to activate DNA damage-induced ubiquitylation for DSB repair by HR (PMID: 23394999).
  4. In section 3.2.1, it is worthwhile to mention that in fission yeast, SUMO-targeted DNA translocase Rrp2 antagonizes with slx8 and displaces SUMOylated TOP2cc to prevent its ubiquitylation and degradation and the subsequent exposure of TOP2-concealed DSB thereby prevent genome instability (PMID: 28552615).
  5. The abbreviations for topoisomerases I and II are TOP1 and TOP2, respectively. TOP I and TOP II are abbreviations for type I (TOP1 and 3 in human and yeast) and II (TOP2 in human and yeast) topoisomerases instead.

This outstanding review is acceptable, and timely for publication in IJMS in 2021.

Author Response

We have uploaded a word file with our detailed response.
